# AN UNCERTAINTY-AWARE FRAMEWORK FOR DATA-EFFICIENT MULTI-VIEW ANIMAL POSE ESTIMATION

## ABSTRACT

Multi-view pose estimation is essential for quantifying animal behavior in scientific research, yet current methods struggle to achieve accurate tracking with limited labeled data and suffer from poor uncertainty estimates. We address these challenges with a comprehensive framework combining novel training and post-processing techniques, and a model distillation procedure that leverages the strengths of these techniques to produce a more efficient and effective pose estimator. Our multi-view transformer (MVT) utilizes pretrained backbones and enables simultaneous processing of information across all views, while a novel patch masking scheme learns robust cross-view correspondences without camera calibration. For calibrated setups, we incorporate geometric consistency through 3D augmentation and a triangulation loss. We extend the existing Ensemble Kalman Smoother (EKS) post-processor to the nonlinear case and enhance uncertainty quantification via a variance inflation technique. Finally, to leverage the scaling properties of the MVT, we design a procedure that exploits improved EKS predictions and uncertainty estimates to generate high-quality pseudo-labels for further training, thereby reducing dependence on manual labels. Our framework components consistently outperform existing methods across three diverse animal species (flies, mice, chickadees), with each component contributing complementary benefits. The result is a practical, uncertainty-aware system for reliable pose estimation that enables downstream behavioral analyses under real-world data constraints.

## 1 INTRODUCTION

Pose estimation has become an indispensable tool for quantifying animal behavior in neuroscience and ethology (Anderson & Perona, 2014; Pereira et al., 2020). Despite ongoing algorithmic advances in deep learning-based pose estimation systems (Mathis et al., 2018; Graving et al., 2019; Pereira et al., 2019; Dunn et al., 2021; Biderman et al., 2024), significant room for improvement remains, particularly in handling complex, multi-camera setups.

Multi-camera pose estimation methods fall into two paradigms depending on their use of camera calibration information. Calibrated methods leverage epipolar geometry and triangulation for superior geometric consistency (Qiu et al., 2019; Zhang et al., 2021b; Dunn et al., 2021; Liao et al., 2024), but require precise camera parameters and break down when cameras are moved (a common occurrence in longitudinal studies or field settings). Uncalibrated methods offer deployment flexibility and robustness to camera movement (Shuai et al., 2022; Zhou et al., 2023), but cannot exploit the powerful geometric constraints that improve pose estimation accuracy when calibration is available. Furthermore, both calibrated and uncalibrated approaches typically process views independently before cross-view fusion of features or heatmaps. This late fusion strategy fails to leverage the rich cross-view correlations available during feature extraction, and limits the ability of these methods to resolve ambiguous keypoints or occlusions that may be clear from alternative viewpoints.

Pose estimation methods, both single- and multi-view, usually also suffer from poorly calibrated uncertainty estimates (Biderman et al., 2024). Although some approaches attempt to improve uncertainty through visibility losses (Doersch et al., 2023) or post-processing techniques using ensembles (Biderman et al., 2024) and Bayesian models (Zhang et al., 2021a), poor uncertainty calibration remains a critical bottleneck for high-precision scientific applications. Another critical constraint in our application domain of animal pose estimation is the scarcity of labeled data. This presents a fun-

damental challenge that current methods inadequately address, as they are typically trained on large human benchmark datasets like COCO (Lin et al., 2014) or Human3.6M (Ionescu et al., 2013) with millions of annotations, while animal behavior studies often have only hundreds of labeled frames.

In this work, we introduce a collection of techniques for improving multi-view pose estimation that addresses these fundamental limitations. Our techniques center on enabling **early cross-view information fusion** by utilizing pretrained transformers that process pixel patches from all views simultaneously, allowing the self-attention mechanism to integrate multi-view information throughout processing rather than through late fusion strategies employed by existing methods. We develop a **multi-view patch masking scheme** that randomly masks pixel patches across views, forcing the model to learn robust cross-view correspondences and effectively utilize information from alternative viewpoints. When camera calibration is available, we implement a **3D augmentation scheme** that maintains geometric consistency across the views and apply a **3D triangulation loss** to predictions from each pair of views. This 3D loss encourages geometric consistency in the predictions themselves and provides a complementary learning signal to the patch masking scheme.

To address the challenge of limited labeled data while improving uncertainty calibration, we refine a recent post-processing algorithm called the "Ensemble Kalman Smoother" (EKS) (Biderman et al., 2024). We implement a nonlinear version and introduce a variance inflation technique that improves uncertainty estimates for both linear and nonlinear cases. These enhancements enable us to identify high-quality pseudo-labels from unlabeled data, which we use to train subsequent networks. This approach effectively transfers the knowledge from the EKS pipeline (which requires training and inference with multiple models) into a single efficient model that achieves comparable performance with dramatically reduced computational overhead.

We demonstrate the effectiveness of each contribution on three diverse multi-view pose estimation datasets spanning different animal models: flies (Karashchuk et al., 2021), mice (Warren et al., 2021), and chickadees (Chettih et al., 2024). The early-fusion multi-view transformer outperforms its single-view counterpart, with patch masking and the 3D loss contributing unique and complementary performance benefits. The variance-inflated nonlinear EKS outperforms the original EKS across all datasets. Finally, we show networks trained from EKS -generated pseudo-labels outperform the original networks, with performance continuing to improve as additional pseudo-labels are incorporated. Together, these techniques offer a collection of simple, model-agnostic approaches that each contribute unique benefits and provide more reliable keypoint tracking for downstream behavioral analyses.

## 2 RELATED WORK

**Multi-view pose estimation.** Multi-view pose estimation has advanced from a two-stage process (independent 2D detection + triangulation) to sophisticated cross-view fusion techniques (Neupane et al., 2024), which can be classified into calibrated approaches (which require known camera parameters) and uncalibrated approaches. Early calibrated approaches relied on CNNs to extract heatmaps from different views, then fused information across views using epipolar geometry (Qiu et al., 2019; Zhang et al., 2021b; Dunn et al., 2021). Epipolar transformers (He et al., 2020) enabled 2D detectors to leverage 3D-aware features through attention mechanisms along epipolar lines. This approach discards information not along the epipolar line from the reference view, which TransFusion (Ma et al., 2021) addressed by introducing the "epipolar field" concept that incorporates information from the entire reference view while maintaining knowledge of epipolar constraints. MVGFormer (Liao et al., 2024) takes a set of initialized queries that encode 3D poses and iteratively refines them using "appearance" and "geometry" modules. Our 3D augmenations and loss, in contrast, are simple to implement and do not require specialized modules, allowing their use with any architecture.

Modern transformer-based approaches exploit the attention mechanism to enable learning implicit cross-view relationships without explicit geometric constraints. The MTF-Transformer (Shuai et al., 2022) pioneered calibration-free multi-view fusion by extracting features from individual views, then fusing features with a transformer head that adjusts pose features using confidence scores to reduce the effect of unreliable 2D detections. MHVformer (Zhou et al., 2023) extends this paradigm with hierarchical multi-view fusion, demonstrating that learned attention mechanisms can effectively replace hand-crafted geometric constraints. Our multi-view transformer and patch masking approaches are further examples of calibration-free techniques, and like the 3D augmentations and

losses, are agnostic to the architecture of the backbone network (as long as it processes sequences of patch embeddings), making them flexible additions to any pose estimation pipeline.

**Pose estimation post-processing.** Post-processing of pose estimation outputs comes in two main categories: single-view (2D) methods and multi-view (3D) methods. Single-view approaches are typically simpler, and include median filters (Mathis et al., 2018; Syeda et al., 2024) and autoencoders (Karashchuk et al., 2021). Multi-view methods offer distinct advantages by leveraging information across camera views, for example with hierarchical Bayesian models (Zhang et al., 2021a) or probabilistic physics-based models (Biderman et al., 2021). Among general multi-view approaches, Anipose (Karashchuk et al., 2021) provides techniques for improving 3D pose estimation through both single-view filters and a triangulation module that integrates temporal and spatial regularization across the whole skeleton. Similarly, GIMBAL (Zhang et al., 2021a) implements a Bayesian model with spatiotemporal constraints over the entire skeleton, using a switching linear dynamical system for temporal smoothness and a hierarchical von Mises-Fisher distribution for spatial constraints on limb lengths and articulation angles. The linear Ensemble Kalman Smoother (EKS) (Biderman et al., 2024) offers a calibration-free approach that implements spatiotemporal constraints over single keypoints and further improves performance using ensembles of networks. Our variance-inflated nonlinear EKS is a calibration-based extension that is more accurate in datasets with large lens distortions and provides improved uncertainty estimates, which are critical for scientific applications.

**Distillation and pseudo-labeling.** Traditional distillation approaches tailored for pose estimation have focused on compressing large teacher networks into smaller student models while maintaining performance (Li et al., 2021; Yang et al., 2023). More recently, pseudo-labeling strategies have emerged as a powerful learning paradigm, where confident predictions on unlabeled data are used to expand the training set (Huang et al., 2023; Li & Lee, 2023), with SelfPose3d a notable example that incorporates geometric consistency in pseudo-label generation (Srivastav et al., 2024). Our work extends this line of research by introducing a novel framework that transfers the knowledge from an ensemble of models processed through multi-view EKS into a single efficient network.

## 3 METHODS

We first discuss our improvements to pose estimation network training: the multi-view vision transformer (MVT), which can be used with any generic vision transformer (VIT) backbone; patch masking, which provides a rich training signal for the cross-view spatial attention of the MVT and does not require camera calibration; and 3D augmentations and loss, which exploit camera calibration information and are agnostic to the pose estimation backbone. Next, we discuss our improvements to the Ensemble Kalman Smoother (EKS) post-processor, which provides improvements over single model predictions. Finally, we discuss how we use pseudo-labels generated by the EKS post-processor to train a single model that is more efficient than EKS and more performant than any single model of the original ensemble.

### 3.1 MULTI-VIEW VISION TRANSFORMER

All of the multi-view pose estimation techniques discussed in the Related Work section employ bespoke architectural elements. While these architectures may provide good performance with enough training data, they do not allow us to easily exploit general pretrained backbones that are useful when training models with a small number of labels. Furthermore, algorithmic simplicity is desirable for our application domain, where users are often experimental labs with little experience maintaining and debugging exotic architectures. Here we propose a simple strategy that allows the model to take advantage of multiple views and is also compatible with generic VIT backbones: rather than process pixel patches from each view independently, we process all patches simultaneously, allowing the standard self-attention mechanism to pool information within and across views.

The standard image VIT (Dosovitskiy et al., 2020) data pipeline (Fig. 1, *top*) starts with a 2D image $\mathbf{x} \in \mathbb{R}^{H \times W \times C}$ (where $H$, $W$, $C$ are height, width, channels) and splits it into 2D patches, each with shape $(P \times P \times C)$, where the patch size $P$ is typically 16. Each patch is reshaped to a vector of length $P^2 C$, and all patches are concatenated into a sequence of the $N$ flattened 2D patches $\mathbf{x}_p \in \mathbb{R}^{N \times (P^2 C)}$, where $N$ is the total number of patches. Each flattened patch $\mathbf{x}_{p,i}$ is mapped with a trainable linear projection to a patch token $\mathbf{z}_i = \mathbf{W}_{\text{proj}} \mathbf{x}_{p,i} + \mathbf{b}_{\text{proj}}$. A standard fixed 1D position encoding $\mathbf{p}_i \in \mathbb{R}^D$ is added to the patch tokens to retain information about the patch location.

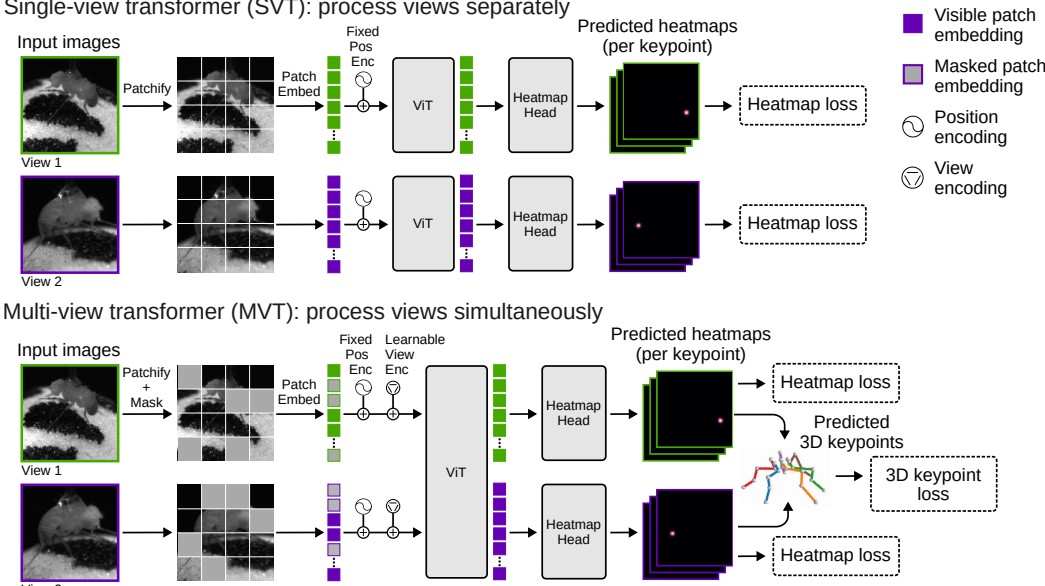

Figure 1: **Multi-view transformer with patch masking and 3D loss.** *Top*: Single-view transformer architecture. Input frames are split into patches, embedded into a latent space, combined with a fixed position encoding, and processed through a Vision Transformer (VIT). Outputs are reshaped and passed to a heatmap head. The model is trained with a mean square error (MSE) loss between predicted and ground truth heatmaps. Multiple views are processed independently. *Bottom*: Multi-view transformer architecture. Pixel patches are randomly masked before patch embedding, then added to a fixed positional and learnable view encodings. A single VIT processes all views simultaneously. The model also produces predicted 3D keypoints using 2D heatmaps and camera calibration, which are compared against ground truth 3D keypoints with an additional MSE loss.

To extend this framework to $V$ camera views (Fig. 1, *bottom*), we apply the same patch embedding pipeline independently to each view $\mathbf{x}^{(v)}$, $v = 1, \ldots, V$. Each pixel patch is projected and enriched with a fixed positional encoding $\mathbf{p}_i$ as before, with an additional learnable view encoding $\mathbf{v}_v$ (inspired by (Ma et al., 2021; Zhou et al., 2023)), resulting in $\tilde{\mathbf{z}}_i^{(v)} = \mathbf{W}_{\text{proj}}\mathbf{x}^{(v,i)} + \mathbf{b}_{\text{proj}} + \mathbf{p}_i + \mathbf{v}_v$. Concatenating all patches from all views forms the joint input sequence $\mathbf{Z}_0 \in \mathbb{R}^{(NV) \times D}$ for the VIT encoder, which produces $\mathbf{Z}_{\text{Enc}} = \text{ViTEncoder}(\mathbf{Z}_0)$. We regroup $\mathbf{Z}_{\text{Enc}}$ by view and reshape to the original 2D patch grids. Following ViTPose (Xu et al., 2022), which showed that VIT backbones retain accuracy with minimal decoders, we employ a lightweight shared upsampling head to map each per-view grid into keypoint heatmaps. See Appendix B.1 for details.

## 3.2 PATCH MASKING

The self-attention of the MVT enables the network to utilize information from multiple views, which is particularly advantageous for handling occlusions. To encourage the model to develop this cross-view reasoning during training, we introduce a pixel space patch masking scheme inspired by the success of masked autoencoders (He et al., 2022) and dropout (Srivastava et al., 2014). Rather than dropping tokens before encoding as in He et al. (2022), we mask patches directly in the input image before patchification, which is more similar to an extreme form of data augmentation mimicking frequent occlusions. We use a training curriculum that starts with a short warmup period where no patches are masked, then increase the ratio of masked patches from 10% to 50% by the end of training. This technique creates gradients that flow through the attention mechanism and encourage cross-view information propagation, which in turn develops internal representations that capture statistical relationships between the different views. We also implement a related view masking technique, where we randomly mask entire views, but find patch masking produces more stable training dynamics (Fig. 6) and better overall performance (Fig. 7). See Appendix B.2 for details.

## 3.3 3D AUGMENTATIONS AND LOSS

The MVT produces a 2D heatmap for each keypoint in each view. Without explicit geometric constraints, it is possible for these individual 2D predictions to be geometrically inconsistent with

each other. If we have access to camera parameters, we can use this additional information to encourage geometric consistency in the outputs. We first take the soft argmax of the 2D heatmaps to get predicted coordinates, following Biderman et al. (2024). Then, for each keypoint, and for each pair of views, we triangulate both the ground truth keypoints and the predictions, and compute the mean square error between the two. The 3D loss is weighted by a hyperparameter, which we set to be the same for both calibrated datasets (Fig. 10). This loss does not require any architectural modifications, and is therefore compatible with a wide range of backbones. It is also complementary to the patch masking scheme, as the 3D loss prevents the model from making predictions that are locally plausible in each view but globally impossible when considered together (Fig. 11).

Data augmentation is a fundamental ingredient to deep learning's success (Mumuni & Mumuni, 2022), especially in the limited-label regime in which we are interested. The 3D loss requires geometrically consistent input images, which precludes applying geometric augmentations like rotation to each view independently. Instead, we triangulate the ground truth labels and augment the 3D poses by translating and scaling in 3D space. The augmented 3D pose is then projected back to individual 2D views. These augmentations do not affect the camera parameters; rather, they are equivalent to keeping the cameras fixed and scaling and translating the subject within the scene. For each view, we then estimate the affine transformation from the original to augmented 2D keypoints, and apply this transformation to the original image (Fig. 8). See Appendix B.3 for details.

## 3.4 Variance-inflated, nonlinear, multi-view Ensemble Kalman Smoother

The linear multi-view Ensemble Kalman Smoother (mvEKS), introduced in Biderman et al. (2024), leverages multi-view constraints by modeling each body part independently, positing that all 2D observations of a given body part should lie in a 3D latent subspace (spatial constraint) and evolve smoothly in time (temporal constraint) (Fig. 3). In this work, we introduce two key advances to the EKS framework. First, we implement a nonlinear version of EKS that utilizes camera calibration information when available. Second, we implement a variance inflation technique that improves both accuracy and uncertainty estimates, both of which we exploit during distillation.

**Linear mvEKS.** Successful post-processing requires identifying which predictions need correction by accurately quantifying uncertainty for each keypoint on each frame. As shown in the original EKS publication (Biderman et al., 2024), the ensemble variance provides a more accurate signal of model uncertainty than network confidence scores. The mvEKS framework integrates this uncertainty signal with spatiotemporal constraints using a probabilistic 'state-space' modeling approach.

We begin with the predictions of an ensemble of $M$ pose estimation networks for a single keypoint across $V$ camera views, $\tilde{\mathbf{X}} \in \mathbb{R}^{T \times 2V \times M}$, where $T$ represents the number of frames, and the factor of 2 accounts for the $(x, y)$ coordinates of the keypoint in each camera view. First, we compute the median and variance across the ensemble dimension to obtain the ensemble median $\mathbf{X}$ and variance $\mathbf{D}$ matrices in $\mathbb{R}^{T \times 2V}$. We then define a state-space model for $\mathbf{X}$ and $\mathbf{D}$ as $\mathbf{z}_t \sim \mathcal{N}(\mathbf{z}_{t-1}, sE_t)$, where the state vector $\mathbf{z} \in \mathbb{R}^3$ captures the 3D nature of the data (Fig. 13), and $s$ is a smoothing parameter scaling the latent dynamics noise matrix $E_t$, for which we implement an automatic hyperparameter selection strategy (Fig. 14). The 3D latent is then linearly mapped to each of $V$ 2D camera views as $\mathbf{x}_t \sim \mathcal{N}(W\mathbf{z}_t + \mu_x, D_t)$, where $W$ is the projection matrix, $\mu_x$ is an offset, and $D_t$ represents the observation uncertainty. Parameter estimation is described in Appendix D.1. We perform inference using standard Kalman filter-smoother recursions.

**Nonlinear mvEKS.** The linear observation model from the previous section works well for datasets with minimal camera distortion. We can partially address larger distortions by increasing the latent space dimensionality (Fig. 13), though this approximation may fail when animals appear near frame edges where distortion is most severe. For calibrated camera setups, we can improve accuracy by replacing mvEKS's linear observations with nonlinear camera projections $f$, yielding $\mathbf{x}_t \sim \mathcal{N}(f(\mathbf{z}_t), D_t)$. The result is a nonlinear Gaussian state space model, and we perform inference using the Dynamax package (Linderman et al., 2025). See Appendix D.2 for details.

**Inflating observed variances for improved uncertainty calibration.** Although ensemble variance provides better uncertainty estimates than individual network confidence scores, ensembles can still be overconfident in certain cases. These overconfident predictions can compromise mvEKS inference and lead to inaccurate posterior variances. To address this limitation, we implement a variance inflation procedure for predictions that are geometrically inconsistent across camera views.

For each observation $\mathbf{x}$ corresponding to a single view, time point, and keypoint, we estimate what this prediction should be (denoted as $\hat{\mathbf{x}}$) based on observations from all other views using either the linear or nonlinear models defined above and an uninformative prior in the latent space. We then assess the discrepancy between $\mathbf{x}$ and $\hat{\mathbf{x}}$ using the Mahalanobis distance, which generalizes the standard $z$-score to multivariate distributions. If this distance exceeds a threshold value (e.g., 5), it indicates a significant mismatch between the observation and predictions from other views, relative to the posterior variance. In such cases, we double the ensemble variance for $\mathbf{x}$, recalculate the Mahalanobis distance, and iterate until the distance falls below the threshold, as increasing the variance progressively reduces the calculated distance. We repeat this procedure for each observation. We then fit mvEKS using the inflated ensemble variances. See Appendix D.3 for details.

### 3.5 DISTILLATION AND PSEUDO-LABELING

Vision transformer performance scales well with data size (Zhai et al., 2022), and we observe the same behavior for our MVT (Fig. 12). However, significantly increasing labeled frames is infeasible for most experimental labs, especially as the annotation burden grows with camera count. To expand the labeled training pool, we leverage our improved mvEKS accuracy and uncertainty estimates through a pseudo-labeling approach. We apply mvEKS to training session videos and compute the summed mvEKS posterior predictive variance across all keypoints and views for each frame. From each video, we retain the $N_f$ frames with lowest variance to filter out low-quality frames where initial estimates lack geometric consistency. Since this pool likely contains many near-duplicate instances, we ensure diversity by performing $k$-means clustering on the 3D poses (from 3D PCA or triangulation) using $N_v$ clusters, then selecting the frame closest to each cluster center. This yields $N_v$ pseudo-labeled frames per video, where the pseudo-labels maintain geometric consistency across views as outputs of mvEKS. After selecting pseudo-labeled frames, we simply combine them with ground truth labels and retrain a single mvEKS-distilled model using the identical training procedure as the initial ensemble members. See Appendix E for details.

## 4 EXPERIMENTAL SETUP

**Datasets.** We demonstrate our contributions on three datasets spanning different animal models (Fig. 2). In "Treadmill Mouse," head-fixed mice run on a circular treadmill while avoiding a moving obstacle (Warren et al., 2021). Seven keypoints are labeled in each of two views, captured at 250 Hz . In "Fly-Anipose," head-fixed flies behave spontaneously on an air-supported ball (Karashchuk et al., 2021). Thirty keypoints are labeled in each of six views, captured at 300 Hz. In "Chickadee," freely moving chickadees engage in seed caching behavior in a large arena (Chettih et al., 2024). Eighteen keypoints are labeled in each of six views, captured at 60 Hz.

**Baselines.** For baselines we compare to our own single-view implementation of ViTPose (Xu et al., 2022), which outperforms ResNet-50 (a widely used backbone in animal pose estimation packages (Mathis et al., 2018; Biderman et al., 2024)) on two of three datasets (Fig. 5). Our baseline MVT implementation uses the same upsampling head as the single-view ViTPose, such that all performance improvements are directly attributable to the early fusion, multi-view processing. For all transformers we use a VIT-S/16 architecture pretrained on ImageNet using DINO (Caron et al., 2021), as we find this backbone compares favorably to other backbones like VIT-B/16 pretrained on ImageNet using either DINO or masked autoencoding (He et al., 2022), and Segment Anything (Kirillov et al., 2023) (Fig. 5). For post-processing, we consider ensembling-based baselines (ensemble median, linear mvEKS) as well as the established triangulation package Anipose (Karashchuk et al., 2021). We describe distillation baselines in more detail in a later section.

**Evaluation.** We train models on 200 frames using three random seeds for the train/validation split. We use the ensemble standard deviation (e.s.d.) for a given keypoint and frame to assess keypoint "difficulty" following Biderman et al. (2024)–a larger e.s.d. across seeds and models means less consensus. We report pixel error as a function of e.s.d., with values at threshold $n$ showing errors for keypoints with e.s.d. $> n$. The leftmost side of each plot shows the error for all keypoints; moving rightward progressively filters to include only more difficult keypoints (those with higher e.s.d.).

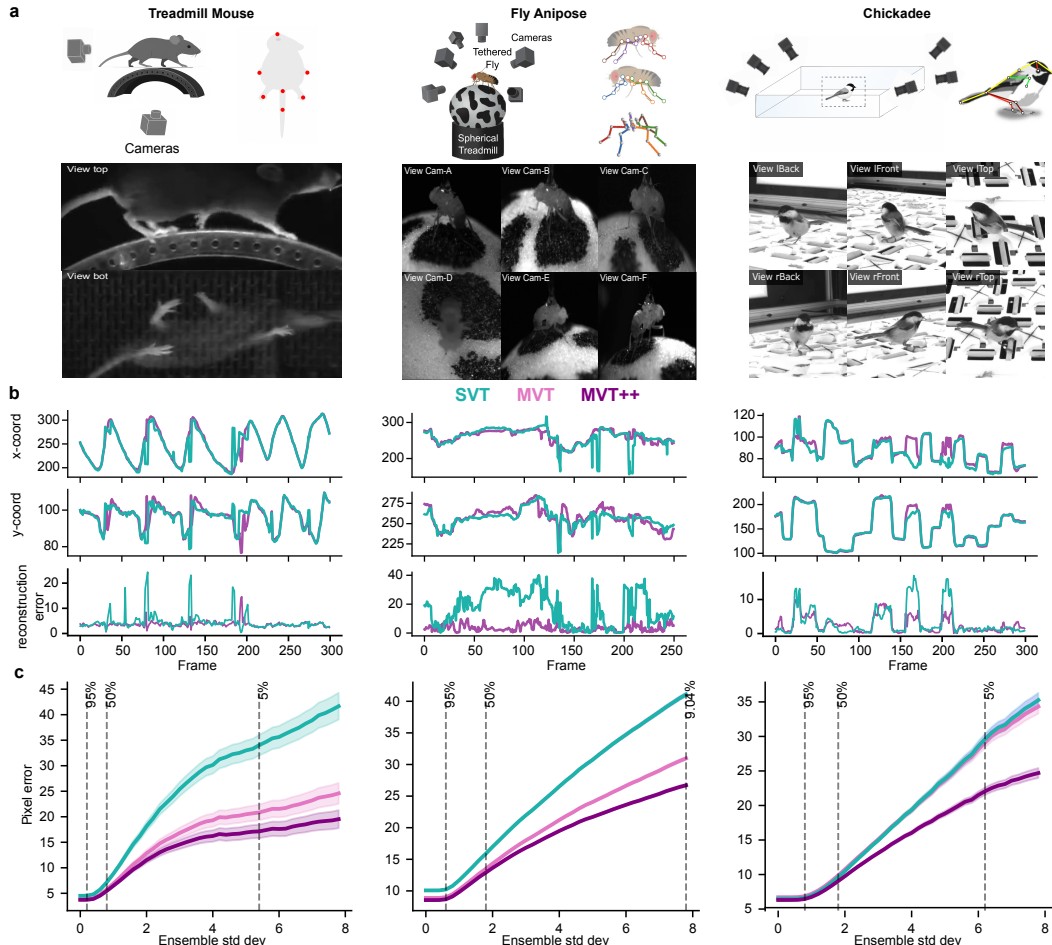

Figure 2: **Multi-view transformer with patch masking and 3D loss improves pose estimation. a:** *Top*: Experimental setup and labeled keypoints. *Bottom*: Example frames for a single instance. **b:** Example traces from single-view transformer (SVT; teal), and the multi-view transformer with patch masking and 3D loss (MVT++; purple; except for Treadmill Mouse, which lacks camera parameters). Bottom panels show 3D reprojection error (using 3D PCA for Treadmill Mouse), indicating more consistent predictions across views for MVT++. **c:** Pixel error as a function of keypoint difficulty (lower is better). Dashed vertical lines indicate the percentage of data used for the pixel error computation. Fly diagram from Karashchuk et al. (2021).

## 5 RESULTS

### 5.1 MULTI-VIEW TRANSFORMERS

The multi-view transformer trained with patch masking and 3D loss (which we refer to as MVT++) consistently outperforms the single-view transformer (SVT) baseline across all datasets, producing smoother predictions with lower reprojection errors (Fig. 2). Ablation experiments reveal the multi-view architecture alone provides substantial gains over SVT on Treadmill Mouse and Fly-Anipose datasets, with comparable performance on Chickadee (Fig. 2). While models using either patch masking or 3D loss individually outperform the base MVT, their combination achieves the best performance across all datasets, demonstrating these components' complementary benefits (Fig. 11).

To verify generalizability across different data regimes, we trained models on subsets of 100 and 400 labeled frames. The MVT++ maintains its advantage over SVT even with only 100 labeled frames, and notably, the performance gap actually increases with more training data, indicating the model scales effectively while remaining robust in limited-data scenarios (Fig. 12). This scaling behavior further motivates our mvEKS-based distillation pipeline.

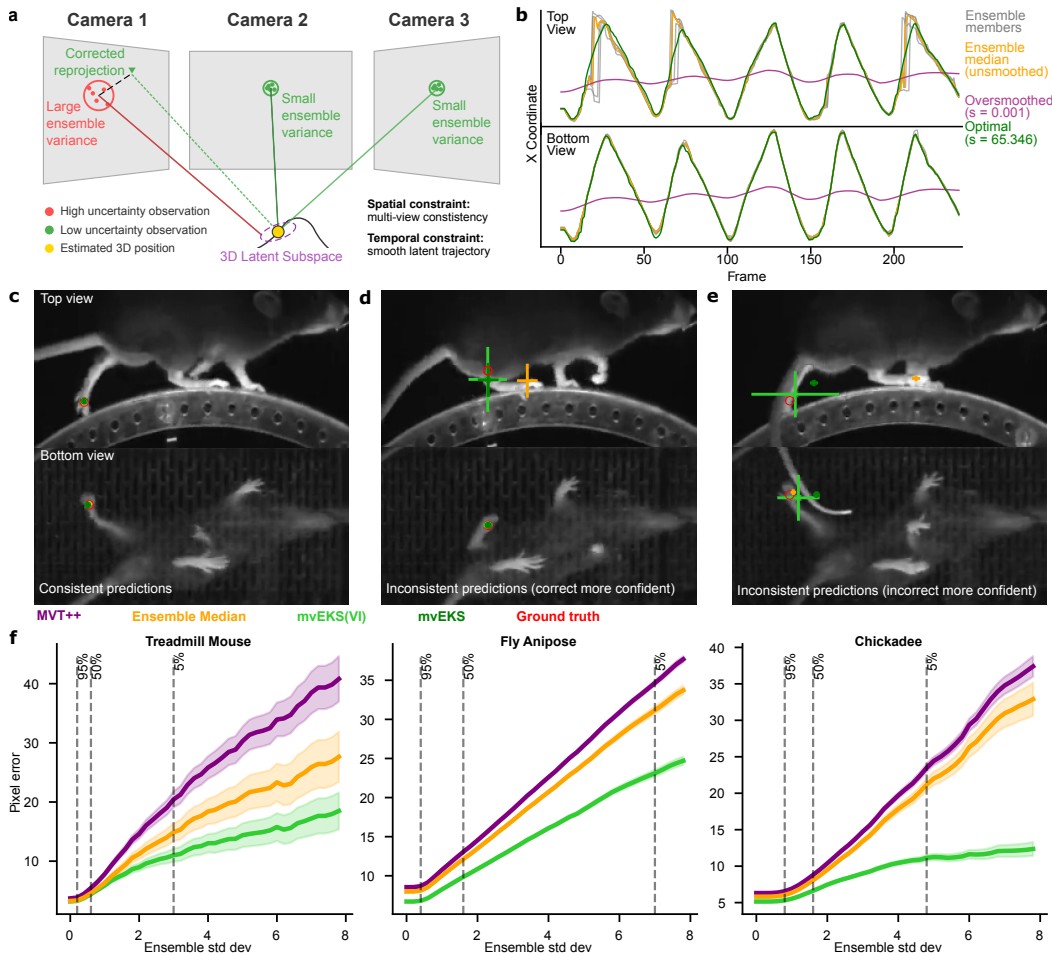

Figure 3: **Multi-view Ensemble Kalman Smoother (mvEKS) improves pose estimation. a:** Keypoints are modeled as projections from a 3D latent that evolves smoothly over time. Low-uncertainty observations from reliable camera views help correct high-uncertainty observations through spatial and temporal constraints. **b:** Traces of a mouse paw from two camera views. The optimal smoothing parameter (green) recovers the true oscillatory motion in the partly occluded Top view, while oversmoothing (purple) distorts the temporal dynamics. **c:** Multi-view observation where predictions are consistent across views, requiring no variance inflation. **d:** Inconsistent predictions across views detected by variance inflation, where the more confident predictions are correct. Orange crosses are ensemble median with ensemble variance; green crosses are corrected predictions from mvEKS with posterior predictive variance. **e:** Inconsistent predictions between views where a highly confident but incorrect prediction in the top view dominates; mvEKS is unable to override the confident error, but the variance inflation procedure adjusts the posterior predictive variance to reflect the remaining uncertainty. **f:** The ensemble median (orange) outperforms individual MVT++ models (purple); nonlinear variance-inflated mvEKS (light green) achieves the best performance. Treadmill mouse (uncalibrated setup) uses linear mvEKS.

## 5.2 MULTI-VIEW EKS

The mvEKS (Fig. 3a,b) provides uncertainty estimates that ideally correlate with prediction errors. When predictions from multiple views align (Fig. 3c), the posterior predictive uncertainty remains low, reflecting high confidence in accurate estimates. When views disagree due to occlusions or ambiguities (Fig. 3d), the variance inflation procedure activates, inflating the posterior predictive variance (green crosses) while simultaneously correcting the prediction. In more challenging scenarios where views disagree but ensemble variance remains low (Fig. 3e), predictions may still be incorrect, resulting in a mismatch between model confidence and actual error. The variance inflation procedure addresses this by increasing the posterior uncertainty, appropriately flagging these predictions as unreliable–even when the prediction cannot be perfectly corrected (Fig. 15).

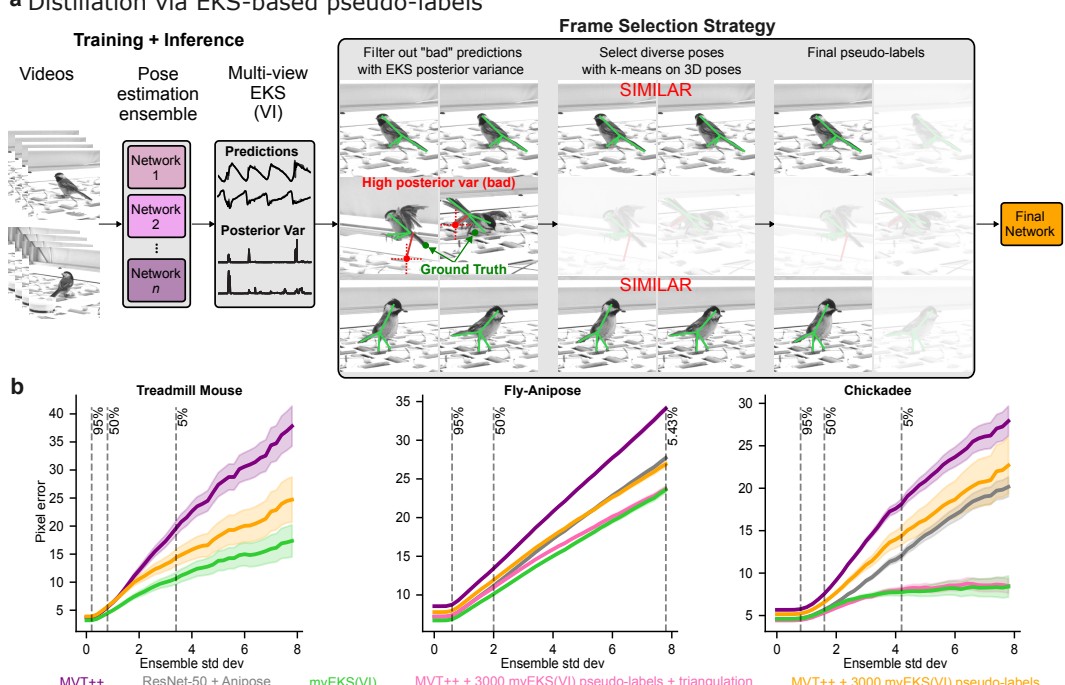

Figure 4: **Pseudo-label-based training of mvEKS improves pose estimation. a:** Schematic of our pseudo-label selection and training procedure. **b:** The distilled MVT+mvEKS model (orange) outperforms initial ensemble member MVT++ models (purple), but does not reach the performance of EKS (green). Enforcing geometric consistency on the distilled model output (pink) brings single-model performance levels equal to that of the full MVT+mvEKS pipeline (green). For calibrated setups, we also compare against the state-of-the-art ResNet-50+Anipose baseline (gray), which performs comparably to our single network distilled model without any post-processing.

We perform an ablation study to demonstrate the impact of different mvEKS components (Fig. 3f). The ensemble median outperforms individual MVT++ ensemble members. For calibrated datasets, our nonlinear mvEKS model achieves further improvements and outperforms Anipose, and dramatically outperforms linear mvEKS (Fig. 16). Note uncalibrated setups cannot use Anipose or nonlinear mvEKS, and linear mvEKS still proves an effective multi-view post processor.

### 5.3 PSEUDO-LABELING

Our pseudo-labeling approach simply and effectively transfers knowledge from the ensemble to a single efficient model (Fig. 4). As expected, mvEKS improves upon the base MVT++ architecture (green vs purple). The MVT++ model trained with high-quality mvEKS-generated pseudo-labels (MVT++PL) achieves improvements over the original MVT++ despite using identical architecture and training procedures (orange vs purple). While the MVT++PL does not reach the full performance of the MVT+mvEKS pipeline, it represents a significant advance in inference efficiency, delivering much better performance than any individual ensemble member while requiring only a single forward pass. Since the MVT++PL model does not enforce geometric consistency during inference, we further enhance it by applying triangulation and reprojection, yielding our best overall performance (pink). This demonstrates our framework successfully captures the knowledge learned by the ensemble, producing a practical single-model solution that approaches the accuracy of our full multi-model pipeline. This performance is enabled by our proposed frame selection method; we find that randomly selecting frames leads to degraded performance (Fig. 17). We note that our MVT++PL model improves upon the current state-of-the-art, Resnet-50+Anipose (gray).

## 6 DISCUSSION

We introduce an uncertainty-aware framework for data-efficient multi-view animal pose estimation comprising three complementary components: improved pose estimation networks (Fig. 2), enhanced post-processing with the variance-inflated, nonlinear Ensemble Kalman Smoother (Fig. 3),

and effective pseudo-label distillation (Fig. 4). We demonstrate how these components work together to improve performance across diverse, data-limited animal pose estimation datasets. Our framework provides benefits at every stage regardless of camera calibration availability, enabling easy adaptation to various experimental setups. Another key strength lies in the framework's simplicity: it requires no large or complex architectures that demand extensive training data, and is readily adaptable to stronger pretrained backbones as they emerge. This framework can be further improved by combining it with other techniques for limited-data regimes, such as domain-specific pretraining (Wang et al., 2025) and semi-supervised learning (Biderman et al., 2024), bringing us closer to simple solutions for accurate multi-view pose estimation in scientific research settings.

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

# A  DATASETS

## A.1  TREADMILL MOUSE

Head-fixed mice run on a circular treadmill while avoiding a moving obstacle (Warren et al., 2021). The treadmill has a transparent floor and a mirror mounted inside at $45°$, allowing a single camera to capture two roughly orthogonal views (side and bottom views via the mirror) at 250 Hz. The camera is positioned at a large distance from the subject ($\sim$1.1 m) to minimize perspective distortion. Frame sizes are $406\times396$ pixels. We split each frame vertically into its respective views in order to make a "multi-camera" dataset. Each view is reshaped during training to $128\times256$ pixels. Seven keypoints on the mouse's body are labeled in each view. The training/test sets consist of 789/253 instances, respectively. We use a 3-dimensional latent space for mvEKS (Fig. 13). We accessed the labeled pose estimation dataset from `https://doi.org/10.6084/m9.figshare.24993315.v1` under the CC-BY 4.0 license.

## A.2  FLY-ANIPOSE

Head-fixed flies behave spontaneously on an air-supported ball, captured by six cameras at 300 Hz (Karashchuk et al., 2021). Frame sizes vary by view, and frames are reshaped during training to $256\times256$ pixels. Thirty keypoints are labeled in each view–five joints on each of six legs.

Our pose estimation models require labels for all views at a given instant in time, and although some of this data is available in the Anipose repository (`https://datadryad.org/dataset/doi:10.5061/dryad.nzs7h44s4`), we took a different approach to ensure a large quantity of high-quality, simultaneously labeled frames. For a subset of sessions in the data repository that contain Anipose predictions, we treat a subset of these predictions as labels for training our own models. We first remove any instance where average 3D reprojection error is $>10$ pixels. When then run $k$-means clustering on the remaining 3D poses (using 25 clusters per session) and select one example per cluster. The training/test sets consist of 377/300 instances, respectively. We use a 3-dimensional latent space for mvEKS (Fig. 13).

## A.3  CHICKADEE

Freely moving chickadees engage in seed caching behavior in a large arena, captured by six cameras at 60 Hz (Chettih et al., 2024). Frame sizes vary by view but are approximately $3000\times1500$ pixels. We created a cropped dataset using the ground truth labels to define a bounding box around the bird, and reshaped the cropped frames to $320\times320$. Each view is reshaped during training to $256\times256$ pixels. Eighteen keypoints on the chickadee's body are labeled in each view. The training/test set consists of 433/143 instances, respectively. We use a 6-dimensional latent space for mvEKS (Fig. 13).

To produce the cropped unlabeled videos for distillation, we implemented a two-stage top-down pose estimation pipeline (Pereira et al., 2020). First, we trained a coarse detector network on full resolution frames downsampled to $256\times256$ pixels to localize the bird within each frame. We then computed a bounding box around the bird in each view, ran inference using a pose estimation model trained specifically on cropped frames, and transformed the resulting predictions back to full-resolution coordinates before applying mvEKS.

# B  MULTI-VIEW POSE ESTIMATION

## B.1  MULTI-VIEW TRANSFORMER

The power of our multi-view transformer (MVT) approach is that it does not require any bespoke or complex architectures, which can require large amounts of data to properly train (Nogueira et al., 2025). Instead, we use encoders from off-the-shelf pretrained transformers combined with simple heatmap heads, which (1) reduces the number of parameters we need to train from scratch; and (2) forces all of the complex cross-view information propagation into the backbone.

We compared a variety of backbones easily accessible through Hugging Face:

- ViT B-16 pretrained on ImageNet with masked autoencoding (He et al., 2022), available at `https://huggingface.co/facebook/vit-mae-base`. The "base" ViT contains ∼80M parameters, which is 4x larger than the ResNet-50 (∼20M parameters). The "16" indicates the model utilizes a patch size of 16×16 pixels.
- ViT B-16 pretrained on ImageNet with DINO (Caron et al., 2021), available at `https://huggingface.co/facebook/dino-vitb16`.
- ViT S-16 pretrained on ImageNet with DINO, available at `https://huggingface.co/facebook/dino-vits16`. The "small" ViT contains ∼20M parameters, on par with ResNet-50.
- ViT B-16 Segment Anything (Kirillov et al., 2023), available at `https://huggingface.co/facebook/sam-vit-base`.

We train the single-view version of each model with three random seeds (Appendix C) and compare to our ResNet-50 baseline pretrained with the Animal AP10K dataset (Yu et al., 2021). The transformers all outperform the ResNet for both Treadmill Mouse and Chickadee, but not for Fly-Anipose (Fig. 5). VIT/S-DINO is the best performing transformer for both Treadmill Mouse and Fly-Anipose, while being the worst for Chickadee. Given these results, we chose VIT/S-DINO for our subsequent experiments due to considerably faster training time than the "base" models (2-3× faster) and lower memory requirements, an important constraint for our domain application where we expect individual labs to be running these models on single consumer-grade GPUs.

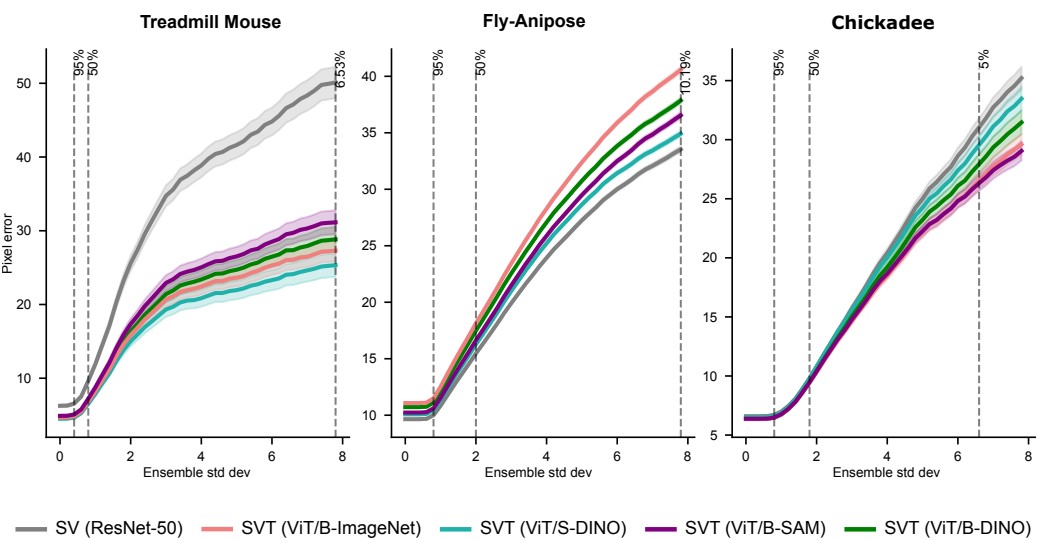

Figure 5: **Comparison of pretrained transformer and ResNet-50 backbones.** VIT/B is a "base" model (∼80M parameters), VIT/S is a "small" model (∼20M parameters); ResNet-50 has ∼20M parameters.

## B.2 PATCH AND VIEW MASKING

The success of masked autoencoding in self-supervised vision transformers (He et al., 2022) inspired us to take a similar approach in the supervised domain of pose estimation. For each labeled instance, we randomly select patches and zero their pixel values before adding position and view encodings. This data augmentation mimics occlusions and forces the transformer to fully exploit cross-view self-attention. We use curriculum learning starting after 700 iterations with 10% patch masking per view, linearly increasing to 50% by iteration 5000.

We also experimented with masking entire views rather than 16×16 patches, using a curriculum that masks single random views after 800 iterations, then two random views after 2900 iterations (if the dataset contains more than two views). However, view masking creates unstable training dynamics with discrete loss jumps (Fig. 6) and provides inferior performance compared to patch masking across datasets (Fig. 7). Additionally, view masking presents generalization challenges for datasets with varying numbers of views, while patch masking applies uniformly to any setup. We therefore adopt patch masking as our default strategy.

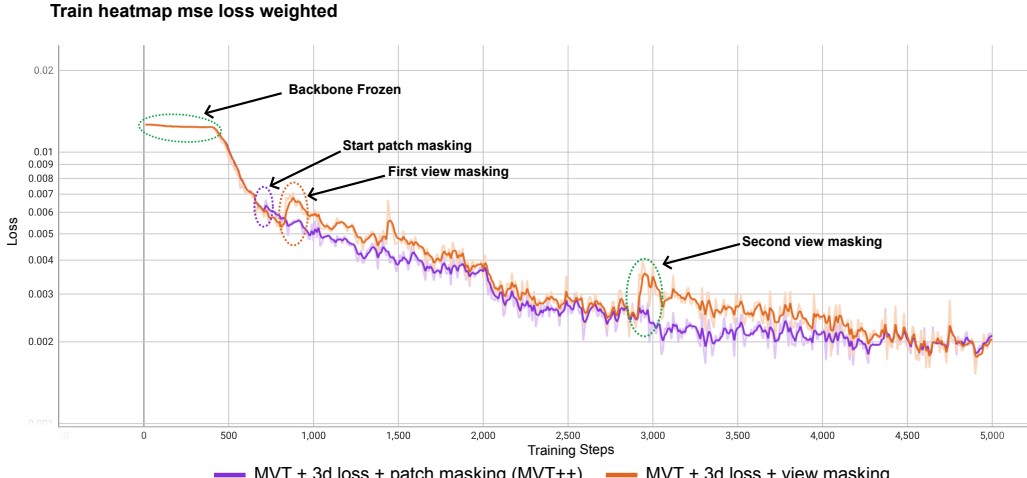

Figure 6: **Patch masking produces smoother training curves than view masking.** Heatmap mean square error loss for the Fly-Anipose dataset (six views) for two different pixel masking strategies.

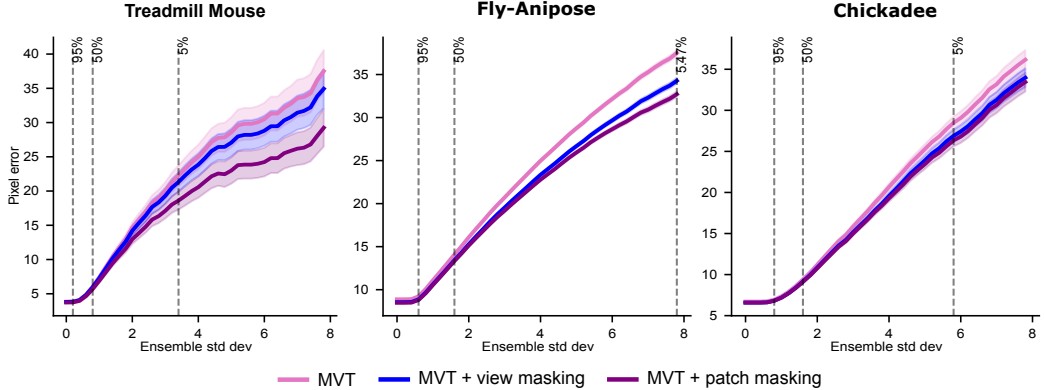

Figure 7: **Patch masking outperforms view masking across datasets.**

## B.3   3D AUGMENTATIONS

We apply the same non-geometric augmentations to all datasets and models using the `imgaug` package with the indicated parameters and probabilities $p$:

- `MotionBlur(k=5, angle=90)`, $p = 0.5$
- `CoarseDropout(p=0.02, size_percent=0.3, per_channel=0.5)`, $p = 0.5$
- `CoarseSalt(p=0.01, size_percent=(0.05, 0.1))`, $p = 0.5$
- `CoarsePepper(p=0.01, size_percent=(0.05, 0.1))`, $p = 0.5$
- `AllChannelsHistogramEqualization()`, $p = 0.1$
- `Emboss(alpha=(0, 0.5), strength=(0.5, 1.5))`, $p = 0.1$

For the single-view models (both ResNets and transformers), we apply additional geometric augmentations, i.e. those which affect the locations of the corresponding keypoints:

- `Affine(rotation=(-25, 25)`, $p = 0.4$
- `ElasticTransformation(alpha=(0, 10), sigma=5)`, $p = 0.5$
- `CropAndPad(percent=(-0.15, 0.15), keep_size=False)`, $p = 0.4$

Standard data augmentation pipelines apply transformations independently to each view, creating problems for multi-view models–particularly those using a 3D consistency loss–that require geo-

metrically consistent augmentations across views for each labeled instance. We therefore implement a 3D data augmentation scheme that maintains geometric consistency.

First, we triangulate 2D ground truth labels using camera parameters to obtain 3D keypoint positions. We randomly scale keypoints by median-centering, multiplying by a random factor drawn from $\mathcal{U}(0.8, 1.2)$, then reapplying the median. Next, we randomly translate keypoints by computing a bounding box in each dimension using the minimum and maximum keypoint coordinates, multiplying its width by a random factor from $\mathcal{U}(-0.25, 0.25)$, and shifting keypoints by the result (such that the shift will be a maximum of 25% of the width of the animal in any direction). Since camera parameters remain fixed, this is equivalent to scaling and translating the subject within the recorded area. To augment images, we reproject the transformed 3D keypoints back to each camera view, estimate view-specific affine transformations between original and augmented labels using OpenCV's `estimateAffinePartial2D`, then apply these transformations to the images (Fig. 8).

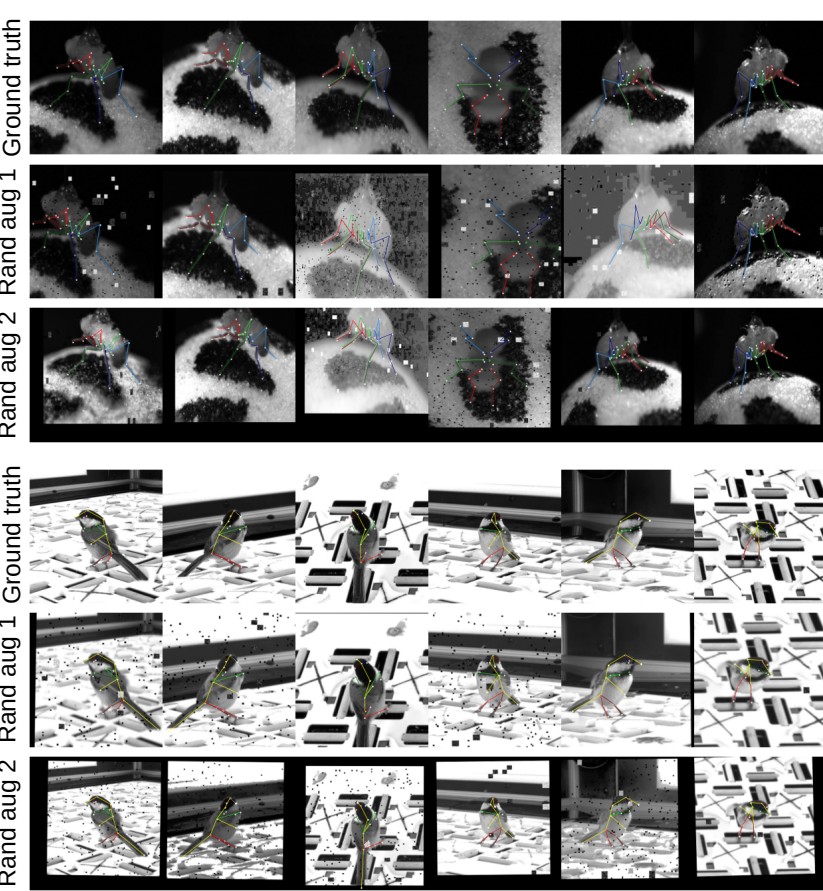

Figure 8: **Example 3D augmentations.** Augmentations for datsets with camera calibration parameters combine scale and translation in the 3D space with view-independent appearance augmentations (e.g., pixel noise and brightness).

On Fly-Anipose and Chickadee datasets, our 3D augmentation performs equivalently to independent-view augmentations (Fig. 9), verifying that the 3D scale and translation hyperparameters are reasonable. The true benefit of this augmentation scheme emerges when paired with the 3D consistency loss, detailed next.

### B.4 3D LOSS

To compute the 3D loss, we first triangulate ground truth 2D labels using camera parameters. For each camera pair, we apply OpenCV's `undistortPoints` and `triangulatePoints` functions, then take the median across all pairs for the final 3D position, following Karashchuk et al.

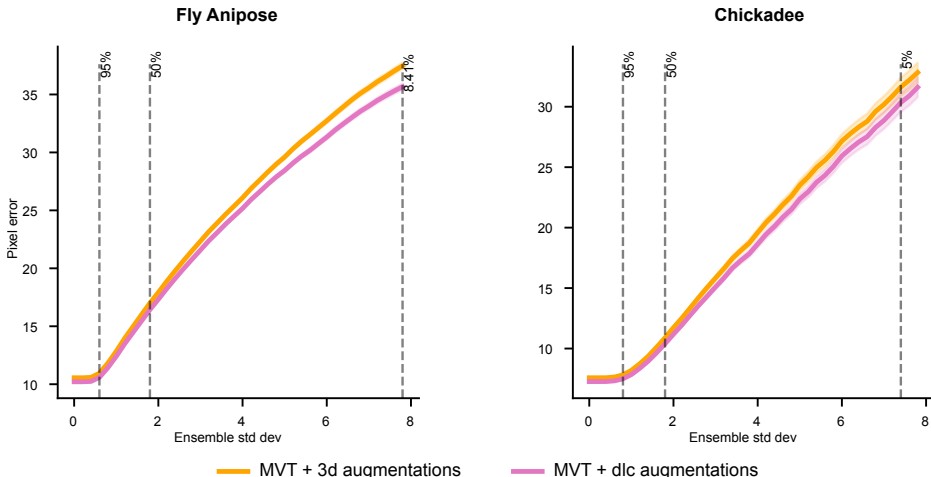

Figure 9: **3D augmentations compare similarly to view-independent augmentations.**

(2021). Ground truth points that move outside the frame boundary during augmentation are marked as `NaN` and excluded from triangulation and loss computation.

Next, we compute the soft argmax (2D spatial expectation) of predicted heatmaps for each keypoint and view. This differentiable operation enables coordinate estimates in downstream losses. Using the same camera parameters, we triangulate the 2D coordinate predictions for each camera pair and compute mean squared error between ground truth 3D keypoints and triangulated predictions for each pair. This forces every view to incorporate information from all other views for all keypoints. The final loss is the mean MSE across all keypoints in the batch, weighted by a scaling factor that balances this loss with the 2D heatmap loss. We find the same scaling factor works well across both datasets ($e^{0.3}$) and therefore use this value for all subsequent experiments (Fig. 10).

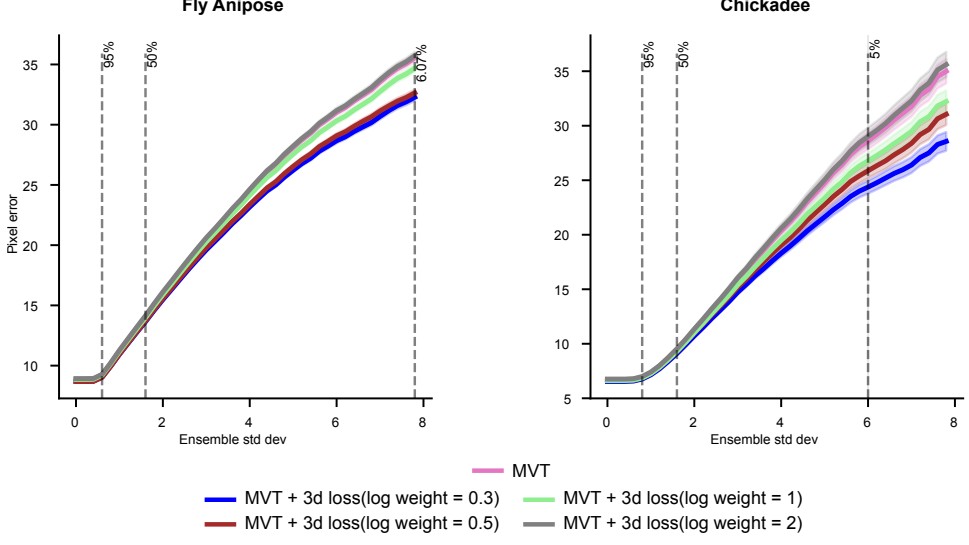

Figure 10: **Comparison of log weight values for 3D loss.**

We find that combining the 3D loss with the patch masking scheme is more powerful than either alone (Fig. 11). View masking creates gradients that flow through the attention mechanisms and cross-view information propagation pathways. This encourages the development of internal representations that capture statistical relationships between different views, regardless of whether those relationships are geometrically motivated. The 3D loss creates gradients that flow back through the

triangulation operation, which means the model receives feedback about how small changes in 2D predictions affect 3D geometric consistency. This encourages the development of internal representations that are naturally geometrically aware. In other words, the view masking ensures the model can handle missing information gracefully, while the 3D loss ensures that the strategies it learns for handling missing information are geometrically sound.

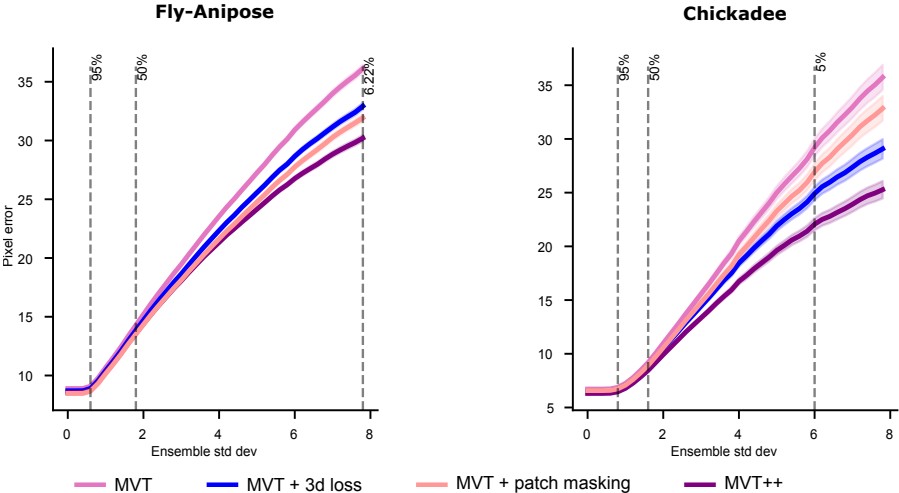

Figure 11: **Patch masking and 3D loss offer complementary performance benefits.**

### B.5 DATA SCALING

All analyses utilize models trained with 200 labeled frames, as this is a reasonable amount of labeled data for a given experimental setup. We also test models using half (100) and twice as many (400) frames to better understand how performance of our different contributions–multi-view transformer, patch masking, and 3D loss–scale with data amounts (Fig. 12). Unsurprisingly, with more labels the performance for all models improves, but more interestingly the full MVT also increases the performance gap over the baselines.

## C POSE ESTIMATION TRAINING

We use Lightning Pose (Biderman et al., 2024) to train supervised pose estimation models on each dataset. Additional details of the model architecture can be found in the original Lightning Pose publication.

For training and inference, we process all camera views simultaneously for each time point. Each batch element comprises one image per camera view (e.g., with six views, a batch size of four contains 24 total images). We use a batch size of eight instances per network.

During training of single-view models, we apply standard image augmentations to labeled frames, including geometric transformations (rotations, crops), color space manipulations (histogram equalization), and kernel filters (motion blur). For data augmentation in multi-view models with camera calibration, see Appendix B.3.

We split the non-test data into 95% for training and 5% for validation. To simulate a limited-data scenario, we randomly select only 100, 200, or `max(400, total_train_frames)` instances from the training set. All evaluations use the model iteration with the lowest validation loss. Different ensemble members use different random seeds for the train/validation split.

We train our models for 5000 iterations using the Adam optimizer (Kingma & Ba, 2014) with an initial learning rate of 0.001, which is halved at iterations 2000, 3000 and 4000. The pretrained backbone remains frozen for the first 400 iterations.

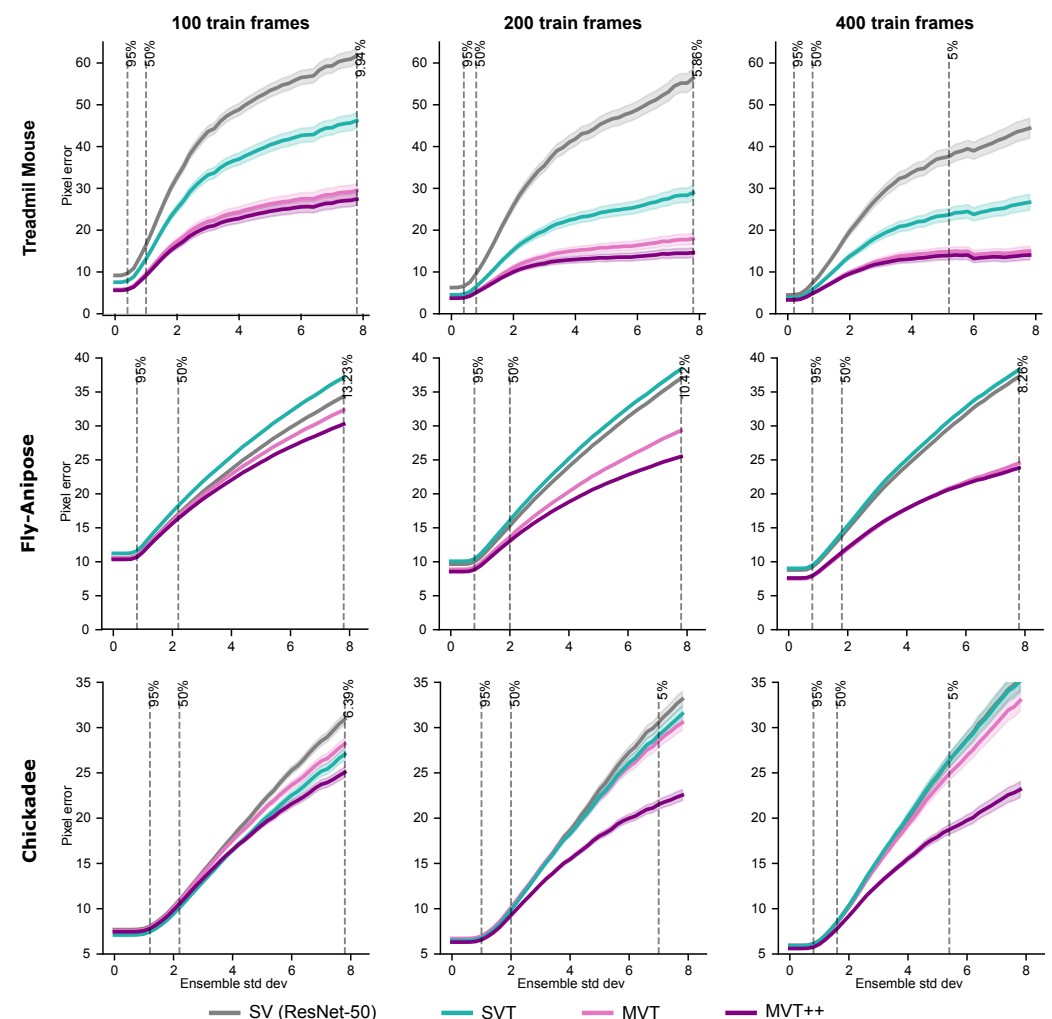

Figure 12: **Model performance as a function of labeled data.** Rightmost results for Fly-Anipose use the maximum of 377 labeled frames.

# D    VARIANCE-INFLATED NONLINEAR ENSEMBLE KALMAN SMOOTHER

We first discuss the PCA-based linear version of mvEKS, describing both parameter initialization and our new automatic smoothing procedure. This version of mvEKS relies on the low-dimensionality of the multi-view data, which we find across all datasets (Fig. 13), and does not require camera calibration. The next section describes the nonlinear mvEKS, which can provide improved performance, especially for setups with larger lens distortion. Finally, we describe in detail the variance inflation procedure that leads to better uncertainty estimates.

## D.1    LINEAR EKS

The linear EKS model described in the main text is

$$
\begin{align}
\mathbf{z}_t &\sim \mathcal{N}(\mathbf{z}_{t-1}, sE_t) \tag{1}\\
\mathbf{x}_t &\sim \mathcal{N}(W\mathbf{z}_t + \mu_x, D_t) \tag{2}
\end{align}
$$

We initialize model parameters by restricting to frames with low ensemble variance and use Principal Component Analysis (PCA) to estimate $W$ and $\mu_x$. We then take temporal differences of the resulting PCA projections and compute their covariance to initialize $E_t$. Finally, we set $D_t$ as a diagonal matrix defined by the ensemble variance at time $t$.

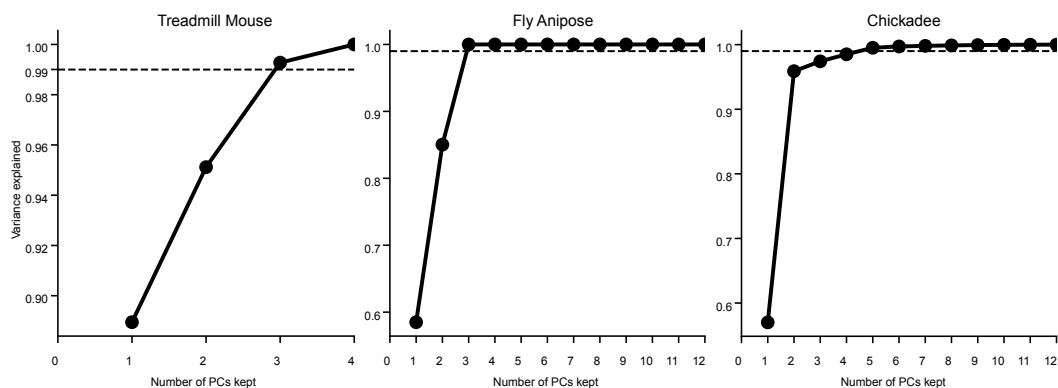

Figure 13: **Multi-view observations are low-dimensional.** Variance explained for increasing numbers of Principal Component dimesnions for each dataset. Three dimensions explain more than 99% of the variance for the mouse and fly datasets, while the chickadee dataset (which includes more camera distortion) requires five dimensions to exceed 99% variance explained.

Selecting the optimal smoothing parameter $s$ in Eq. 1 is crucial: too large leads to undersmoothing, while too small causes oversmoothing (Fig. 3b). The optimal parameter occupies a well-defined minimum in the log-likelihood loss landscape (Fig. 14a) and must be learned from data, as it varies substantially across keypoints and videos (Fig. 14b). To perform automatic tuning of this parameter, we use the Adam optimizer implemented in the `optax` package with learning rate set to $0.25$.

We initialize $s$ using the standard deviation of the temporal differences of the initial PCA projections, which we have found to often lie near the minimum of the log-likelihood loss function. To improve the computational efficiency, we implemented a version of EKS that parallelizes over keypoints using the JAX library (Bradbury et al., 2018).

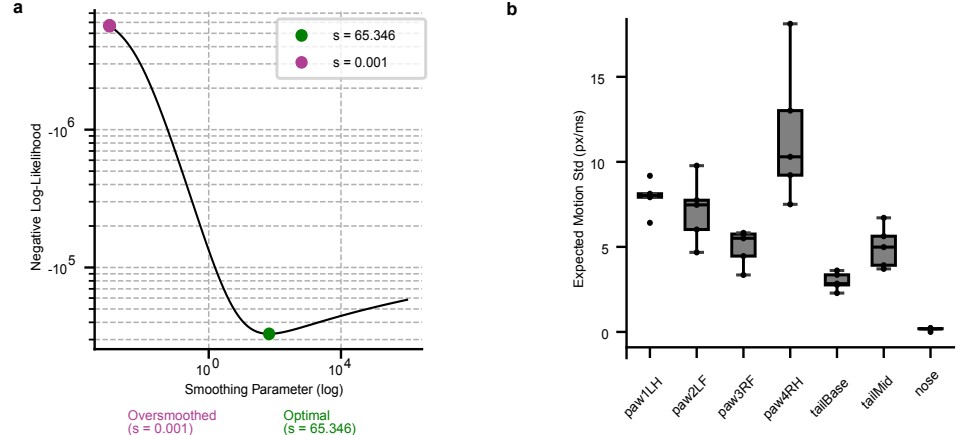

Figure 14: **Automatic smoothing for mvEKS. a**: Log-likelihood as a function of mvEKS smoothing parameter shows a single well-defined optimum. **b**: Expected motion magnitude varies by keypoint in the Treadmill Mouse dataset: the nose exhibits low expected motion (high temporal consistency), while the paws show greater expected variability in position across frames (pixels per millisecond).

## D.2  NONLINEAR MVEKS

The nonlinear mvEKS model uses camera calibration parameters to map the latent state $\mathbf{z}_t$ in Eq. 1 to the observations $\mathbf{x}_t^j$ in view $j$:

$$\mathbf{x}_t^j \;\sim\; \mathcal{N}\left(f_j(\mathbf{z}_t), D_t^j\right), \tag{3}$$

where $f_j$ represents a standard camera model with radial and tangential distortion (Hartley & Zisserman, 2003). To clearly describe this transformation, we adopt the following notation: "world" coordinates $\mathbf{z}_t$ are denoted as $(\tilde{X}, \tilde{Y}, \tilde{Z})$, and final "image" coordinates $\mathbf{x}_t^j$ for a single view as $(u, v)$. What follows describes the coordinate transformation for a single view; we apply this transformation to the world coordinates for every view (each with its own parameters) and concatenate to arrive at the final observations $\mathbf{x}_t$.

**Step 1: World to camera coordinates.** We first transform world coordinates to 3D camera-based coordinates using the camera extrinsics: a rotation matrix $R$ and translation vector $\mathbf{t}$ that define the camera's position relative to the world coordinate system:

$$\begin{pmatrix} X \\ Y \\ Z \end{pmatrix} = R \begin{pmatrix} \tilde{X} \\ \tilde{Y} \\ \tilde{Z} \end{pmatrix} + \mathbf{t}. \tag{4}$$

**Step 2: Perspective projection.** The camera coordinates are then normalized by dividing by the depth $Z$, which performs perspective projection onto the image plane:

$$x = X/Z \tag{5}$$
$$y = Y/Z. \tag{6}$$

**Step 3: Distortion correction.** Real cameras introduce distortion that must be modeled. We apply two types:

*Radial distortion* accounts for lens curvature effects based on distance $r$ from the image center:

$$r^2 \;=\; x^2 + y^2 \tag{7}$$
$$d_r \;=\; 1 + k_1 r^2 + k_2 r^4, \tag{8}$$

where $k_1$ and $k_2$ are calibrated distortion coefficients.

*Tangential distortion* corrects for lens misalignment:

$$x_t \;=\; 2p_1 xy + p_2(r^2 + 2x^2) \tag{9}$$
$$y_t \;=\; 2p_2 xy + p_1(r^2 + 2y^2). \tag{10}$$

The combined distorted coordinates are:

$$x_d \;=\; x \cdot d_r + x_t \tag{11}$$
$$y_d \;=\; y \cdot d_r + y_t. \tag{12}$$

**Step 4: Pixel coordinates.** Finally, we apply the intrinsic camera matrix containing focal lengths $(f_x, f_y)$ and optical centers $(c_x, c_y)$ to convert to pixel coordinates:

$$\begin{pmatrix} u \\ v \\ 1 \end{pmatrix} = \begin{pmatrix} f_x & 0 & c_x \\ 0 & f_y & c_y \\ 0 & 0 & 1 \end{pmatrix} \begin{pmatrix} x_d \\ y_d \\ 1 \end{pmatrix}. \tag{13}$$

The nonlinear function $f_j$ thus combines the camera extrinsics $(R, \mathbf{t})$, distortion parameters $(k_1, k_2, p_1, p_2)$, and camera intrinsics $(f_x, f_y, c_x, c_y)$. For both Fly-Anipose and Chickadee datasets these parameters are obtained using standard camera calibration techniques, as described in Karashchuk et al. (2021).

### D.3 VARIANCE INFLATION

We start with the general case of computing the Mahalanobis distance with an uninformative prior; the next section details how to apply this to multi-view pose estimation for outlier detection.

Let $\mathbf{x} \in \mathbb{R}^n$ be a vector of observations. We describe these observations with a standard linear latent variable model:

$$p(\mathbf{x}|\mathbf{z}) = \mathcal{N}(\mathbf{x}|W\mathbf{z} + \mu_x, D), \tag{14}$$

where $\mathbf{z} \in \mathbb{R}^d$ is a set of unobserved latent variables. For simplicity we will assume $D$ is a diagonal matrix (note that with this assumption Eq. 14 becomes the generative model of Factor Analysis (Bishop, 2006)).

The posterior distribution of the latents given the observations is

$$
\begin{aligned}
p(\mathbf{z}|\mathbf{x}) &= \mathcal{N}(\mathbf{z}|BW^TD^{-1}(\mathbf{x} - \mu_x), B) && (15) \\
&= \mathcal{N}(\mathbf{z}|\mu_{z|x}, B), && (16)
\end{aligned}
$$

where $B = (I + W^TD^{-1}W)^{-1}$ if the prior on $\mathbf{z}$ is $\mathcal{N}(0, I)$. However, this is a strong assumption, and instead we can use an uninformative prior where $\mathbf{z} \sim \lim_{\sigma \to \infty} \mathcal{N}(0, \sigma I)$; this results in

$$
B = (W^TD^{-1}W)^{-1}. \tag{17}
$$

Next we will consider the posterior predictive distribution $p(\mathbf{x}'|\mathbf{x})$, which describes the distribution of a new observation $\mathbf{x}'$ given the observed data $\mathbf{x}$:

$$
p(\mathbf{x}'|\mathbf{x}) = \mathcal{N}(\mathbf{x}'|W\mu_{z|x} + \mu_x, D + WBW^T). \tag{18}
$$

We can now use this information to compute the distance between the original observation $\mathbf{x}$ and the posterior predictive distribution, which is essentially measuring the reprojection error of the observation scaled by a covariance matrix. If we define $Q = D + WBW^T$ (the covariance of the posterior predictive distribution), then the Mahalanobis distance is computed as

$$
d_{\text{Maha}} = (\mathbf{x} - \mathbf{x}')^T Q^{-1} (\mathbf{x} - \mathbf{x}'). \tag{19}
$$

For more information on these derivations see Bishop's Pattern Recognition and Machine Learning textbook (Bishop, 2006).

**Mahalanobis distance for multi-view pose estimation.** We would like to use the Mahalanobis distance to measure reprojection errors of multi-view pose estimates; this distance can then provide a metric for the quality of pose estimates without ground truth labels.

This metric will be computed one body part at a time (across all camera views). Assume we have $V$ camera views. For a given instant in time there will be an (x, y) prediction across all $V$ views; stack these values into a single vector $\mathbf{x} = [x_1, y_1, \dots, x_V, y_V] \in \mathbb{R}^{2V}$. This $2V$-dimensional vector represents a point in 3D space, so we can model it with the linear latent variable model of Eq. 14. The unique approach here is that instead of learning a single covariance matrix $D$ for all observations, we will utilize observed ensemble variances that change from one observation to the next.

Now, if we consider the posterior predictive variance as defined in Eq. 18, the resulting $Q$ would represent a single, joint measure of discrepancy across all camera views simultaneously. However, we would like to compute the posterior predictive variance $Q^v$ for a single view $v$ that incorporates information from the other views. Conditioning on the observations from the other views is straightforward in a linear model. Let us define $D^v \in \mathbb{R}^{2 \times 2}$ as the diagonal block of observed variances in $D$ for view $v$; and $W^v \in \mathbb{R}^{2 \times 3}$ as the two rows of the loading matrix $W$ that correspond to view $v$. Then

$$
Q^v = D^v + W^v B (W^v)^T. \tag{20}
$$

Finally, if we define $\mathbf{x}^v = [x_v, y_v]$ to be the observations for view $v$, then

$$
d_{\text{Maha}}^v = (\mathbf{x}^v - \mathbf{x}^{v'})^T Q^{-1} (\mathbf{x}^v - \mathbf{x}^{v'}). \tag{21}
$$

This distance $d_{\text{Maha}}^v$ is the one we compare against a threshold to determine if the observed ensemble variances in $D^v$ should be increased (in our case, scaled by a factor of two).

**Special case: two camera views.** The above is a general procedure that becomes more robust as the number of camera views $V$ increases. However, with only two views ($V = 2$), we face potential indeterminacy issues. When predictions from both views are inconsistent yet each has low ensemble variance, it is impossible to determine which view (if either) is correct. Therefore, in the two-view case, if the Mahalanobis distance exceeds our threshold for one view, we inflate the variance in both views rather than trying to identify the problematic view (Fig. 15).

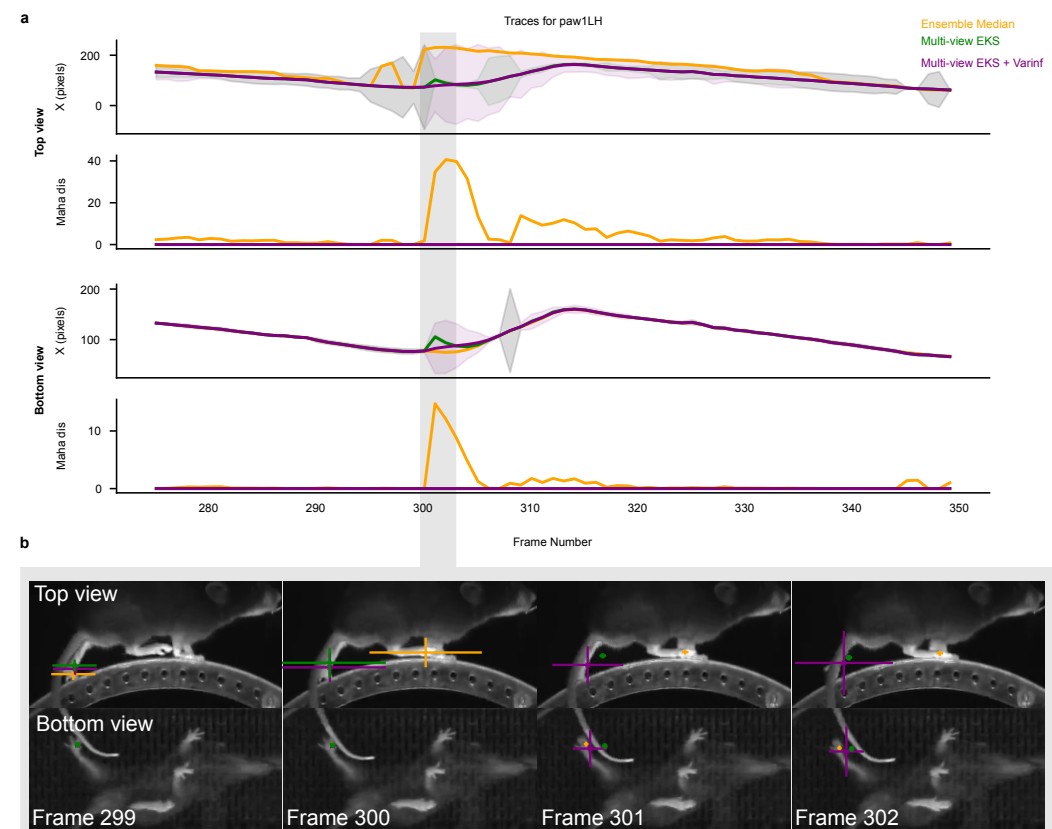

Figure 15: **Variance inflation resolves cross-view inconsistencies in multi-view pose estimation. a**: Example traces from the ensemble median, (linear) mvEKS, and mvEKS with variance inflation for a single keypoint in Treadmill Mouse (left hind paw) across top and bottom views of a held-out video. A segment with an occluded paw is shaded in gray. Frames with high Mahalanobis distance indicate low confidence or disagreement across camera views. mvEKS improves temporal smoothness and cross-view coherence compared to the ensemble median, while variance inflation further resolves residual inconsistencies by penalizing overconfident predictions and enforcing agreement across views. **b**: Sequence of frames (299–302) corresponding to the occlusion region in **a**. The ensemble median exhibits cross-view disagreement under occlusion. mvEKS shows improved consistency, and variance-inflated mvEKS fully aligns top and bottom view predictions through increased uncertainty regularization.

# E   ENSEMBLING AND DISTILLATION

Our distillation pipeline processes mvEKS predictions consisting of 2D keypoint coordinates accompanied by uncertainty estimates. The pipeline employs a two-stage selection procedure designed to ensure both *prediction quality* and *pose diversity*.

**Stage 0: Inference and post-processing.** After model training is complete (here, we use ensembles of three networks, each with a different train/validation data split), inference is run on videos from the training set. The predictions are then post-processed with mvEKS.

**Stage 1: Quality-based filtering.** Frames are first filtered based on uncertainty estimates. When available, we utilize the posterior variance from the mvEKS; otherwise, ensemble variance is applied (for example, when using the ensemble median rather than mvEKS as a baseline). To quantify instance-level confidence, we compute the maximum variance across all keypoints and views:

$$\sigma_{\max}^2 = \max_{k,v} \left\{ \sigma_{x,kv}^2, \sigma_{y,kv}^2 \right\}, \tag{22}$$

where $k$ indexes keypoints and $v$ indexes views. Frames are ranked by $\sigma_{\max}^2$, and the $N_f$ frames with the lowest values are retained, prioritizing predictions with the lowest variance. We set $N_f = 450$ for Fly-Anipose (where videos are short, ~600 frames), $N_f = 21,000$ for Treadmill Mouse (where videos are long, ~30,000 frames) and $N_f = 1200$ for Chickadee (where videos are ~1800 frames).

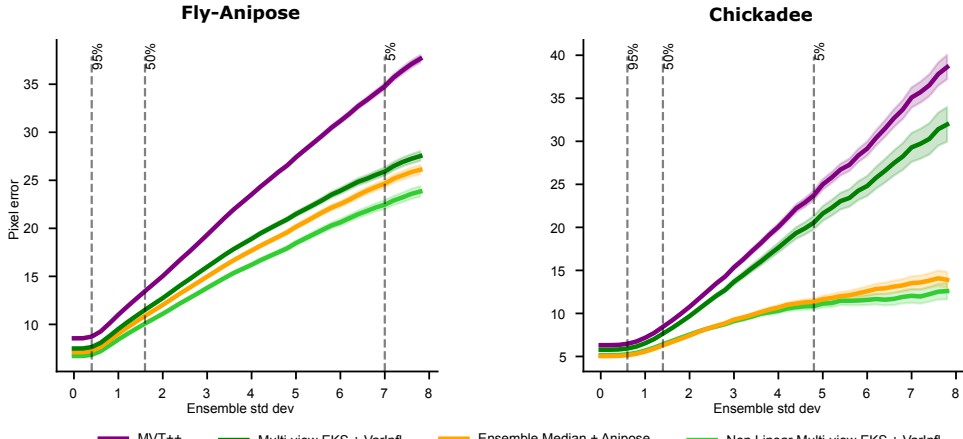

Figure 16: **Calibrated post-processor comparison.** Nonlinear mvEKS (light green) outperforms both linear mvEKS (dark green) and Anipose run on the ensemble median (i.e., same input as the mvEKS models; orange).

**Stage 2: Diversity-based filtering.** The subset of high-confidence frames is then subjected to clustering in 3D pose space to promote diversity. When camera parameters are available, 3D poses are obtained via triangulation; otherwise, PCA-based projection is used. $k$-means clustering is performed to identify representative poses. For each cluster $j$, we select the frame closest to its center:

$$i_j^* = \arg\min_{i \in \mathcal{C}_j} \left\| \mathbf{x}_i^{3D} - \mathbf{c}_j \right\|^2, \tag{23}$$

with $\mathcal{C}_j$ denoting the set of frames assigned to cluster $j$.

The final selected frames are then converted into pseudo-labels using the original 2D mvEKS predictions. Empirically, we find that incorporating variance inflation—by leveraging the posterior variance from mvEKS—provides a more reliable quality measure compared to ensemble-based variance (Fig. 15), leading to improved frame selection.

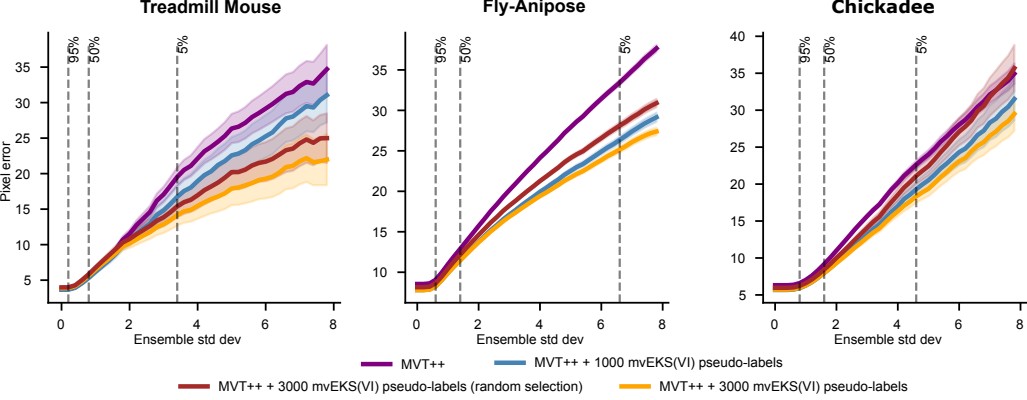

Figure 17: **Pseudo-label based distillation pipeline.** Targeted frame selection outperforms random selection.

# F    REBUTTAL APPENDIX

## F.1    3D PAIRWISE PROJECTION LOSS

To encourage geometric consistency across camera views, we define a **pairwise 3D loss** that penalizes discrepancies between triangulated 3D predictions and the ground-truth 3D keypoints.

For each batch element $b \in \{1, \ldots, B\}$, camera pair $p = (v_1, v_2)$ drawn from the set of all camera pairs $P$, and keypoint $k \in \{1, \ldots, K\}$, we first compute the predicted 3D keypoint via triangulation:

$$\hat{\mathbf{y}}_{b,p,k}^{(3D)} = \text{Triangulate}\big(\hat{\mathbf{u}}_{b,k}^{(v_1)}, \; \hat{\mathbf{u}}_{b,k}^{(v_2)}; \; \Pi_{v_1}, \Pi_{v_2}\big), \tag{24}$$

where $\hat{\mathbf{u}}_{b,k}^{(v)} \in \mathbb{R}^2$ are the predicted 2D keypoint coordinates for view $v$, obtained via the differentiable soft-argmax over heatmaps, and $\Pi_v$ is the camera projection matrix for view $v$. Ground-truth 3D keypoints $\mathbf{y}_{b,k}^{(3D)}$ are computed by triangulating annotated 2D labels and taking the median across all valid camera pairs, following (Karashchuk et al., 2021). Missing or out-of-frame keypoints are excluded from the loss via a binary visibility mask $m_{b,p,k} \in \{0, 1\}$, which is only equal to 1 if keypoints are visible in both views. Some keypoints may not be visible if augmentation has moved the ground truth keypoints outside of the frame. The loss is then defined as:

$$\mathcal{L}_{\text{pairwise-3D}} = w \cdot \frac{\sum\limits_{b=1}^{B} \sum\limits_{p \in P} \sum\limits_{k=1}^{K} m_{b,p,k} \big\| \hat{\mathbf{y}}_{b,p,k}^{(3D)} - \mathbf{y}_{b,k}^{(3D)} \big\|_2}{\sum\limits_{b,p,k} m_{b,p,k}}, \tag{25}$$

where $w = e^{\log\text{-weight}}$ is a fixed hyperparameter that balances the aforementioned term with the 2D heatmap loss. This formulation averages the Euclidean distance between predicted and ground-truth 3D coordinates across all valid pairs and keypoints, enforcing global geometric coherence among views.

## F.2    PATCH MASKING FORMULATION

**Masking function.** Let $\mathbf{I}_{b,v} \in \mathbb{R}^{C \times H \times W}$ denote an image from view $v$ in batch $b$, divided into $N$ non-overlapping patches $\{\mathbf{x}_{b,v,i}\}_{i=1}^{N}$ of size $P \times P$. For each view $v$, a subset of patches $\mathcal{M}_{b,v}$ is randomly selected for masking:

$$\tilde{\mathbf{x}}_{b,v,i} = \begin{cases} \mathbf{x}_{b,v,i}, & \text{if } i \notin \mathcal{M}_{b,v}, \\ \mathbf{0}, & \text{if } i \in \mathcal{M}_{b,v}, \end{cases} \quad \text{with } \frac{|\mathcal{M}_{b,v}|}{N} = r_t. \tag{26}$$

Masked images $\tilde{\mathbf{I}}_{b,v}$ are then projected and combined with fixed positional encodings $p_i$ and learnable view embeddings $v_v$:

$$\tilde{\mathbf{z}}_{b,v,i} = W_{\text{proj}} \tilde{\mathbf{x}}_{b,v,i} + b_{\text{proj}} + p_i + v_v. \tag{27}$$

This formulation encourages the transformer to propagate information across views, leveraging unmasked views to infer masked content.

**Curriculum schedule.** We gradually increase the masking ratio $r_t$ during training to stabilize optimization:

$$r_t = \begin{cases} 0, & t < t_{\text{init}}, \\ r_{\text{init}} + \frac{t - t_{\text{init}}}{t_{\text{final}} - t_{\text{init}}}(r_{\text{final}} - r_{\text{init}}), & t_{\text{init}} \le t \le t_{\text{final}}, \\ r_{\text{final}}, & t > t_{\text{final}}. \end{cases} \tag{28}$$

We typically set $r_{\text{init}} = 0.1$, $r_{\text{final}} = 0.5$, $t_{\text{init}} = 700$, and $t_{\text{final}} = 5000$ training steps. This curriculum enables early stabilization followed by increasingly challenging cross-view reasoning.

## F.3    RELATED WORK ON MASKING STRATEGIES

**Masked modeling and reconstruction-based masking.** Masking has been extensively studied in self-supervised vision learning, most notably through masked image modeling (MIM). Masked Autoencoders (MAE; (He et al., 2022)) introduced token-level masking in the transformer feature space, reconstructing missing patches using a lightweight decoder. Variants such as BEiT (Bao et al., 2022), SimMIM (Xie et al., 2022), and iBOT (Zhou et al., 2022) further validate reconstruction-driven masking as an effective means of learning visual representations. Most masking schemes aim to reconstruct missing patches from within the same image.

**Pixel-space masking and occlusion-based regularization.** A complementary line of work applies masking directly in pixel space. Random Erasing (Zhong et al., 2020) removes contiguous regions of the input image to improve robustness to occlusion. Subsequent work showed that Vision Transformers, in particular, benefit substantially from such erasing: Naseer et al. (2021) demonstrated that ViTs exhibit strong robustness and long-range integration under patch erasure, while Park & Kim (2022) showed that spatial masking applied before patchification improves generalization and optimization stability in ViTs. These findings indicate that pixel-space masking can serve as a structural inductive bias rather than solely a reconstruction pretext task.

**Comparison to our approach.** Our strategy is motivated by these insights but differs in two crucial ways. First, unlike MAE-style methods, we do not train a decoder nor reconstruct masked content; rather, we train a pose estimation head that maps (possibly masked) tokens to full keypoint heatmaps. Second, unlike generic pixel-space occlusion augmentations, our goal is not only robustness but also cross-view correspondence learning. By masking patches in pixel space before patchification in a synchronized multi-view setting, we force the transformer to reason about occluded or missing regions in one view using complementary cues from other views. This turns masking into a form of structural cross-view supervision, promoting view-consistent representations and improving the handling of occlusions and partial visibility across views (Fig. 22). This converts masking from a traditional "self-supervised pretext" into a structural supervision signal that drives cross-view attention and view-to-view consistency.

## F.4 PATCH MASKING HYPERPARAMETER ANALYSIS

We conducted a systematic hyperparameter analysis of the patch masking technique by varying the patch mask ratio (0.1, 0.25, 0.5, 0.75) and masking start iteration (0, 700, 1000, 2000). As expected, performance varies across hyperparameter settings, with optimal values showing some dataset dependence. For Treadmill-Mouse and Chickadee, increasing the mask ratio continues to improve performance, whereas the opposite is true for Fly-Anipose. The iteration at which patch masking starts seems less important than the masking ratio, but starting the masking at the first iteration is generally worse. Importantly, our original hyperparameter choices (ratio=0.5, start iteration=700) demonstrate consistent and reasonably strong performance across all evaluated datasets, suggesting these values provide a robust default configuration. Having robust defaults that work across datasets is especially relevant for our application domain, as we strive to make tools that work out of the box for experimental labs with as little fine-tuning as possible.

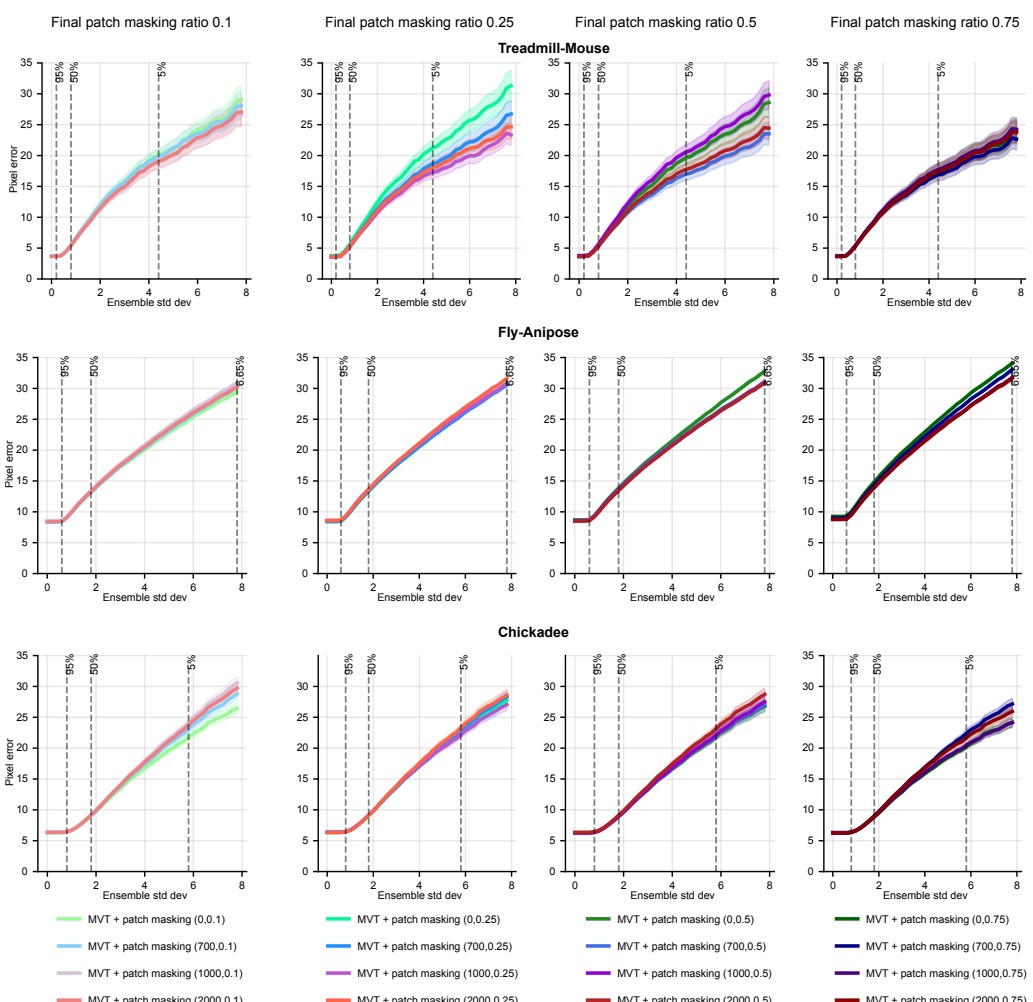

Figure 18: **Patch masking ablations.** Pixel error as a function of ensemble standard deviation for various patch masking hyperparameters (start iteration and mask ratio).

## F.5 MVT TIMING TESTS

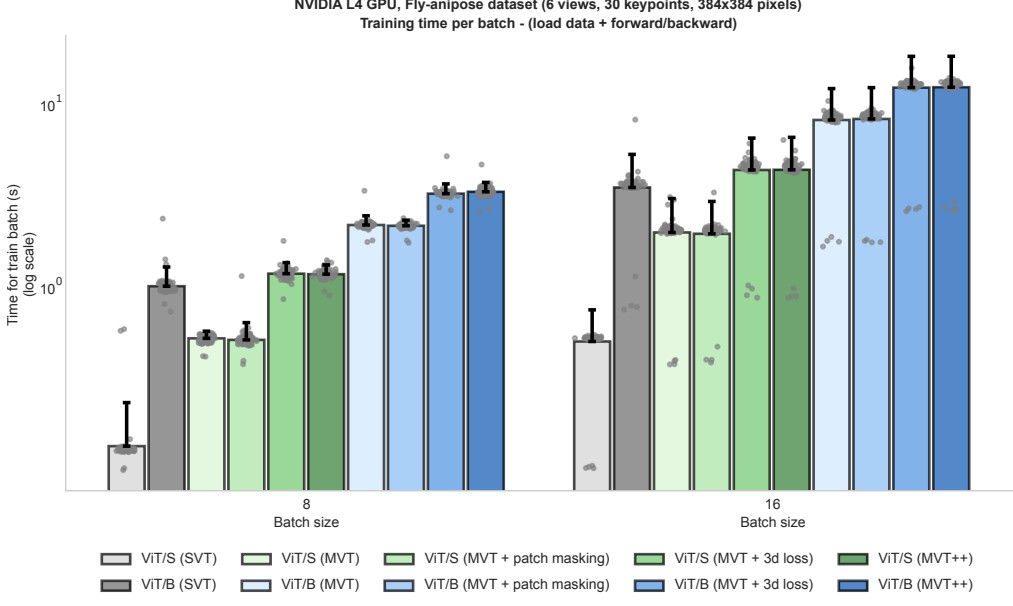

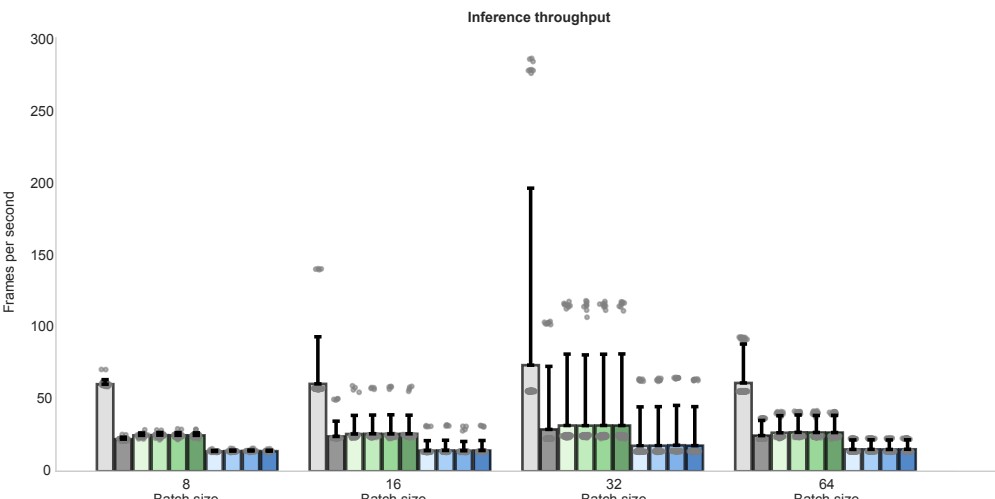

Figure 19: **Training time per batch and inference throughput.** Top panel: each bar depicts the mean batch processing time (in seconds) and 95% CI over $n = 100$ batches, with each of the batches overlaid as a point, using a log-scale spacing for the y-axis. For robustness, 50 warm-up batches were processed before recording any measurements. Bars are grouped by batch size along the x-axis (8/16). Bottom panel: each bar depicts the mean frames-per-second with 95% CI over $n = 100$ batches, and every batch measurement is overlaid as a gray point on a linear y-axis. Bars are grouped by batch size (8/16/32/64). For robustness, 50 warm-up batches were processed before recording any measurements.

## F.6 EKS BASELINES

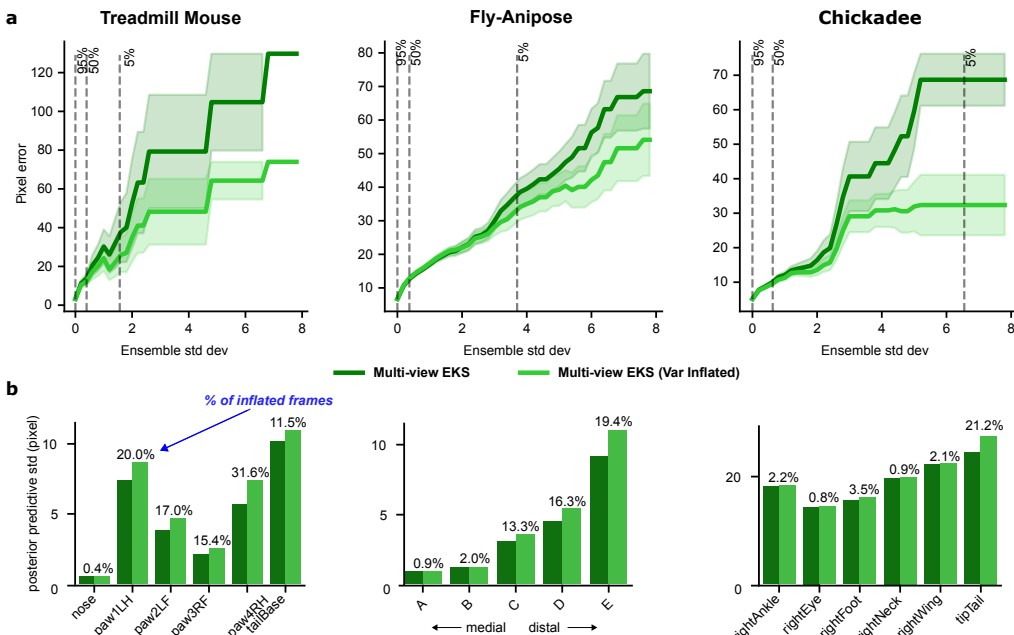

Figure 20: **Multi-view EKS with variance inflation procedure improves Multi-view EKS baseline. a.** Pixel error versus ensemble standard deviation comparison between mvEKS (green) and mvEKS with variance inflation (lightgreen). **b.** Posterior predictive standard deviation for mvEKS with and without variance-inflation for a subset of points across three datasets. Percentages indicate proportions of frames where variance is inflated. Across datasets, we find that challenging distal keypoints such as paws and limbs require variance inflation more frequently, likely due to higher variability and occlusions. In contrast, more medial and stable keypoints like the nose and head tend to produce consistent predictions across views and therefore require less adjustment

## F.7 NUMBER OF VIEWS ABLATION

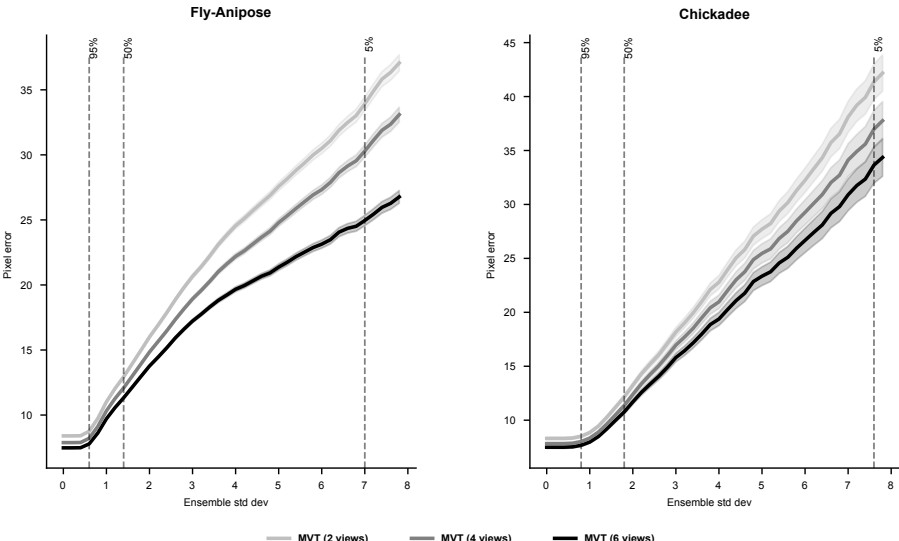

Figure 21: **The accuracy of MVT increases with the number of observed views.** The MVT model was trained on the Fly-Anipose and Chickadee datasets (both six-view). We varied the number of views used during training to either two, four, or six views (three random seeds per condition). The two views used for the two-view condition were a random subset of the four views, which in turn were a random subset of the six available views. All models were evaluated on the same shared two-view subset for 2D keypoint prediction. This design isolates the finding that increasing the number of views used during training significantly improves the MVT model's performance, as evaluation is held constant across all conditions.

### F.8 Occlusion handling

We designed a systematic experiment where we completely mask an entire view from test instances and evaluate the model's ability to predict keypoints in the occluded view using information from remaining views. This process is repeated across all views and test instances to provide comprehensive occlusion performance metrics. The single-view transformer (SVT) serves as our baseline: since it processes views independently, complete view occlusion should result in poor predictions due to lack of access to complementary information. We compare SVT against MVT, MVT with patch masking, and (for calibrated datasets) MVT with 3D loss, and the full MVT++. Any improvement over SVT demonstrates successful utilization of cross-view information to compensate for occlusions. Patch masking provides substantial improvements in occlusion handling (Fig. 22), reducing SVT occlusion errors by 71% on Fly-Anipose, 42% on Chickadee, and 31% on Treadmill Mouse. These results demonstrate that our cross-view patch masking strategy effectively teaches the model to leverage complementary information across views.

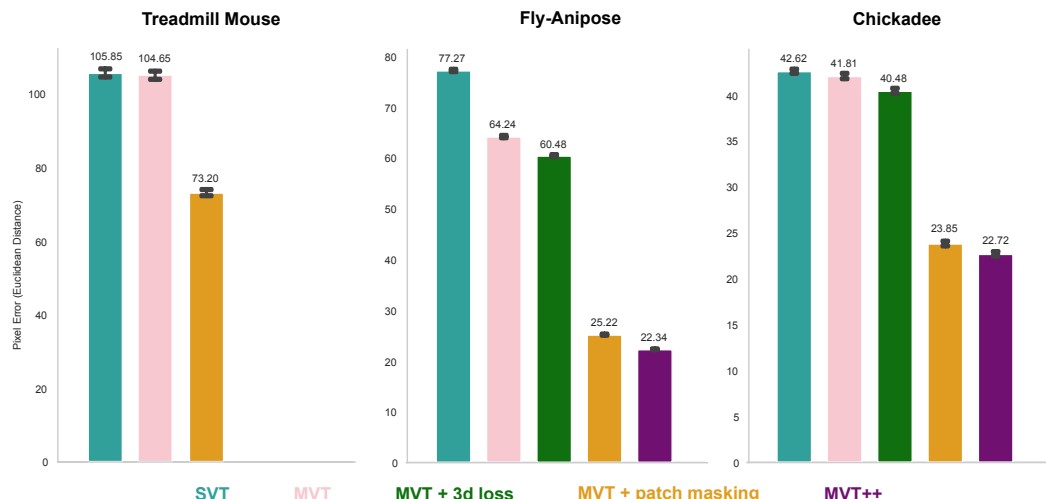

Figure 22: **Model performance on occluded keypoints.**

This figure demonstrates the superior performance of MVT++ in tracking rapid, small-magnitude angular movements. MVT++'s fusion of information from multiple cameras inherently overcomes view-specific challenges, such as keypoint occlusion (e.g., paw or body self-occlusion), which degrade the performance of the single-view baseline, SVT (Single-View Transformer).

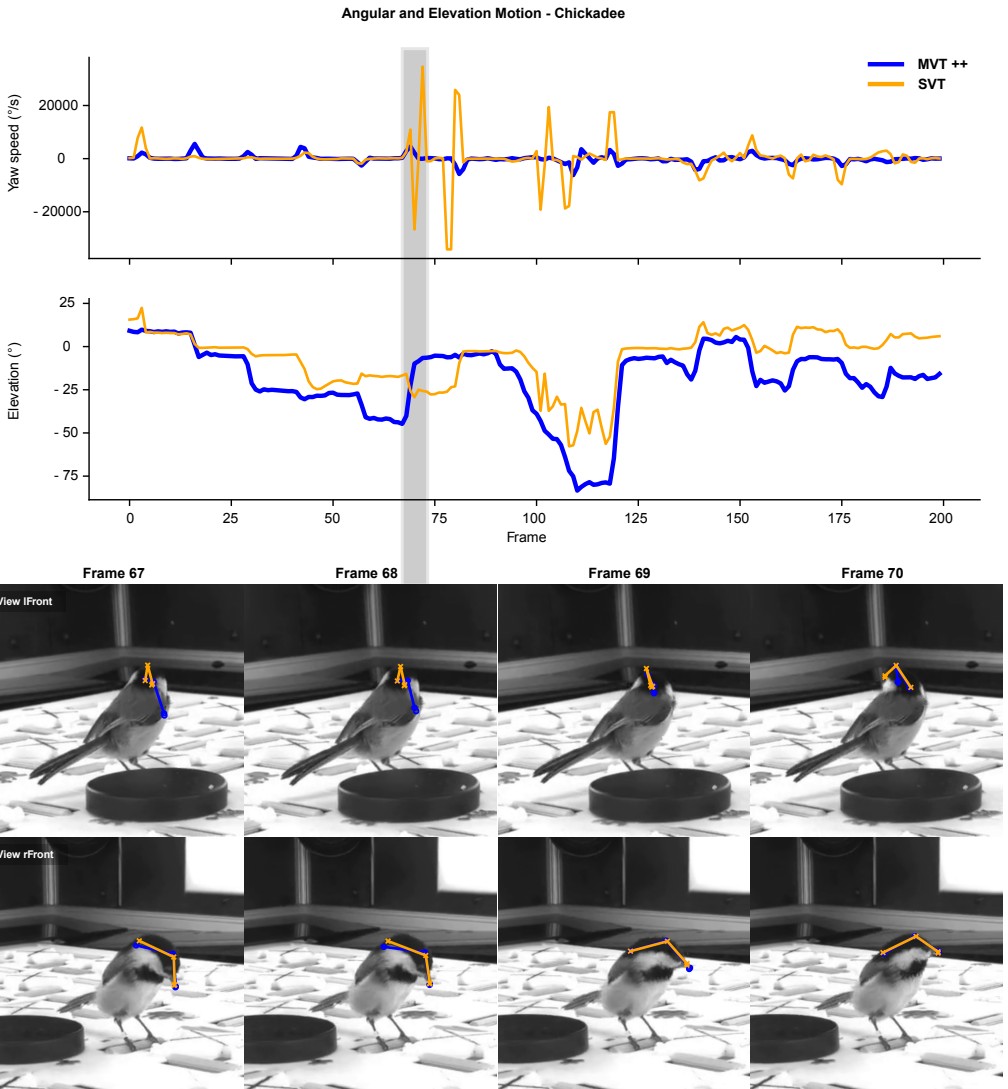

Figure 23: **MVT++ captures high-frequency head direction dynamics. Top:** Time-series of estimated Yaw Speed (°/s) and Elevation (°) shown for 200 video frames. MVT++ (blue) exhibits stable, localized motion estimates, while SVT (orange) shows noisy behavior, especially in the Yaw Speed trace. The shaded region highlights a rapid head direction change between Frame 67 and 70. MVT++ correctly registers a sharp, localized spike in Yaw Speed corresponding to this event, whereas SVT fails to register the key movement. **Bottom:** Frames 67 to 70 illustrate the head movement from two camera views (*iFront* and *rFront*). The overlaid keypoints visually confirm the fast, low-magnitude rotation. The stability and responsiveness of the MVT++ trace during this critical event underscore its ability to generalize across views and maintain temporal coherence during complex, high-frequency biological motions, validating its enhanced accuracy over single-view methods.

## F.9 BACKBONE ARCHITECTURE MVT + SVT

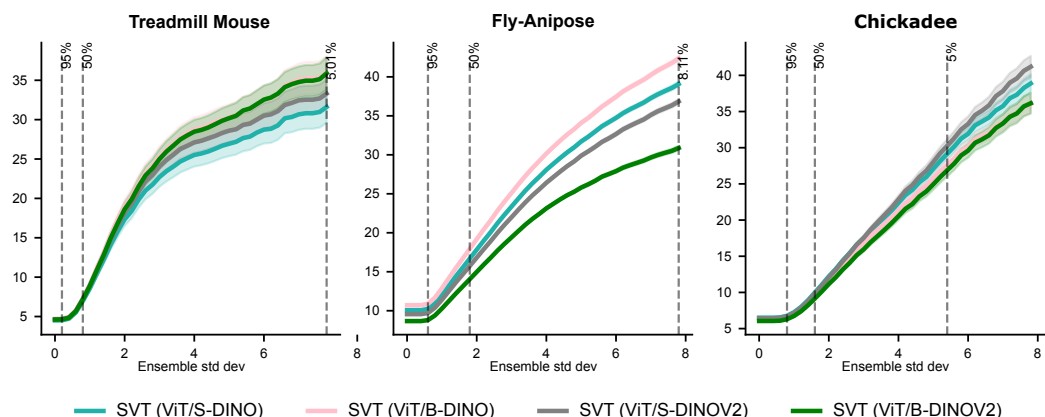

Figure 24: **Comparison of pretrained transformer backbones for SVT.**

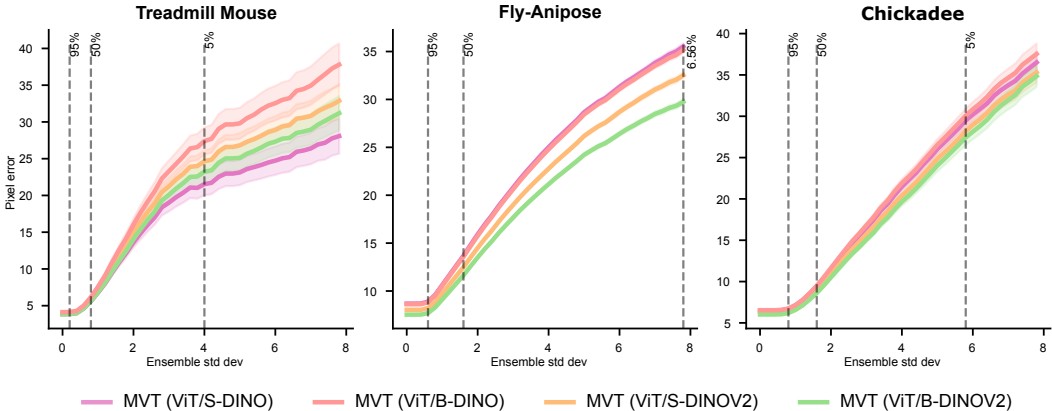

Figure 25: **Comparison of pretrained transformer backbones for MVT.**

## F.10 3D DATA AUGMENTATION

Our data augmentation pipeline comprises two components: (1) a 2D component applying non-geometric augmentations (blur, noise, etc.) independently to each view, and (2) a 3D geometric component applied simultaneously across all views. The 3D geometric augmentations maintain consistency across views through the following process. We first triangulate 2D ground truth keypoints from each view to obtain 3D coordinates, then compute a 3D bounding box spanning the minimum and maximum keypoint locations in each dimension. The keypoints are centered by subtracting the median keypoint value in each dimension. The *scale* augmentation (Fig. 26, top row) samples a random scalar from $[0.8, 1.2]$ and multiplies each centered keypoint by this value. This resizes the 3D bounding box while preserving the subject's aspect ratio and maintaining the centered position. The *translate* augmentation (Fig. 26, middle row) samples random scalars from $[-0.25, 0.25]$ for each dimension, then multiplies these by the corresponding bounding box dimension length. This produces translation amounts that scale appropriately with subject size.

The bottom two rows of Fig. 26 illustrate example augmentations, showing how aspect ratios remain constant while translation vectors vary by dimension. After applying 3D transformations, the augmented keypoints are projected back to each 2D view. Finally, images are transformed to match the geometric changes by fitting an affine transformation between original and augmented 2D keypoints, then applying this transformation to the original images to generate the augmented images. See Fig. 9 for examples using real data.

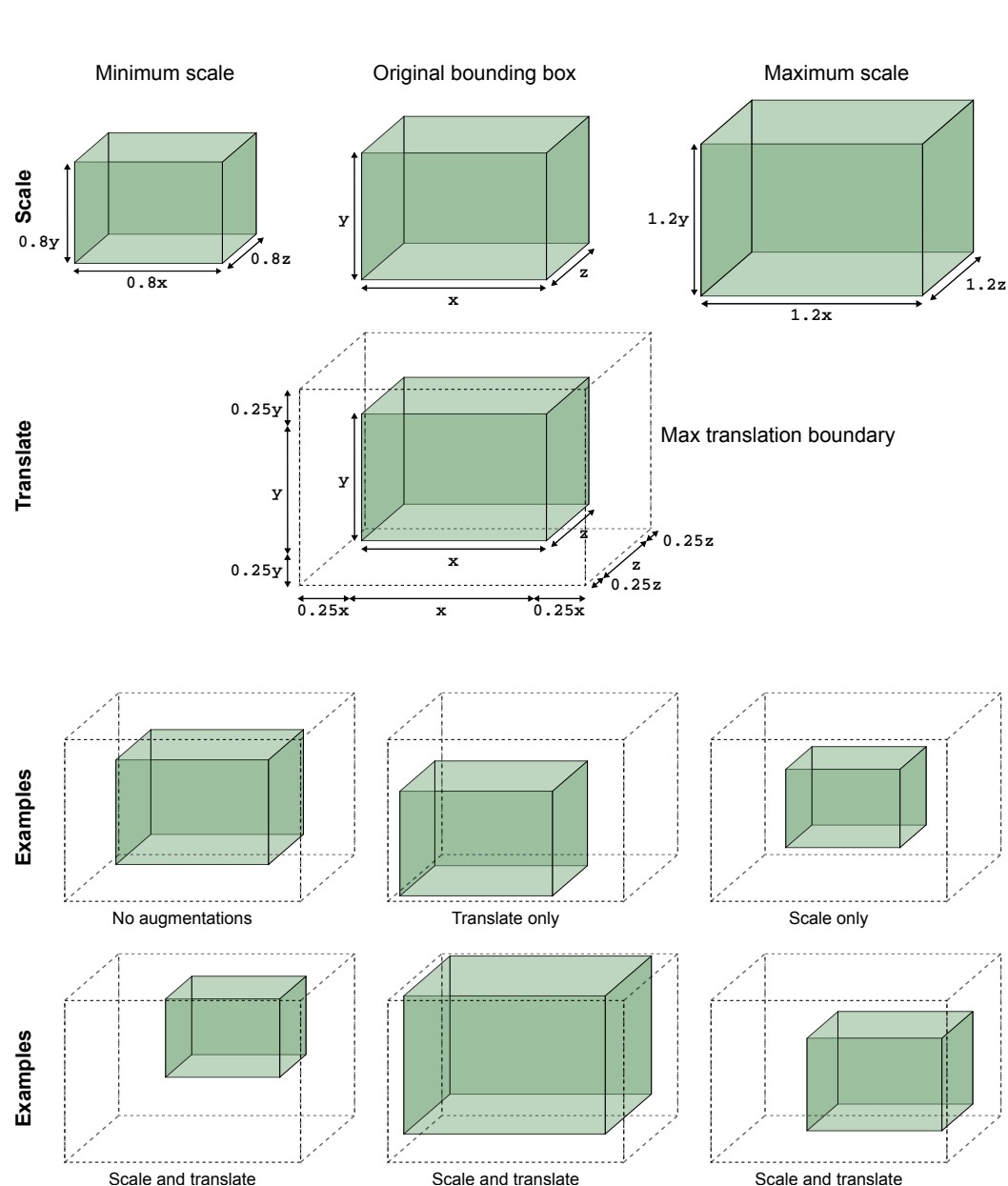

Figure 26: **Illustration of 3D geometric data augmentations.**

## F.11 DANNCE Baseline

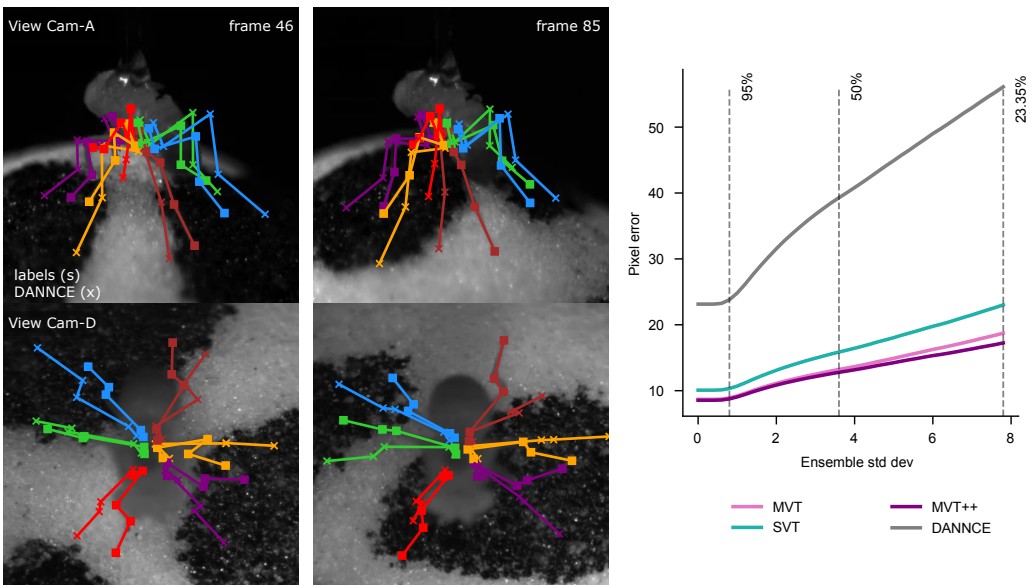

Figure 27: **DANNCE trained on 200 labeled frames.**

