# OpenReview forum: "An uncertainty-aware framework for data-efficient multi-view animal pose estimation"
_ICLR.cc/2026/Conference — ICLR 2026 Conference Desk Rejected Submission_

### Official Review · Reviewer_rzyx · 2025-10-25

**Soundness:** 3
**Presentation:** 3
**Contribution:** 2
**Rating:** 4
**Confidence:** 3

**Summary:**

Multi-view animal pose estimation methods have several major limitations:
(1) they typically process each camera view separately and only fuse information late, which wastes cross-view geometric information;
(2) they provide poorly calibrated uncertainty; and
(3) animal datasets often have very few labeled frames, so the amount of supervised data is limited.
To address these challenges, the authors propose a new architecture, an uncertainty-aware post-processing method, and a pseudo-labeling pipeline:
Architecture. The authors introduce a multi-view vision transformer (MVT) that jointly processes pixel patches from all camera views, enabling early fusion through attention. They also add a multi-view patch masking scheme that randomly masks patches in some views.

Uncertainty estimation. The authors extend Ensemble Kalman Smoothing (EKS) into a variance-inflated and nonlinear versions.

Pseudo-labeling / “distillation.” The authors then use these uncertainty estimates provided by EKS to select high-confidence frames: they keep frames with low predicted variance, filter near-duplicates via clustering, and treat those as pseudo-labels. They combine these pseudo-labels with the ground-truth annotations and retrain a single model. The practical benefit is that, at inference time, you can run one model instead of running an ensemble and post-processing with mvEKS every time, while still getting the gains from the ensemble+mvEKS pipeline.

The authors evaluate on three animal datasets (treadmill mouse, fly, chickadee) and show that the proposed MVT achieves lower pixel error than single-view baselines, and patch masking further improves performance under limited labels. For the mvEKS / variance inflation part and for the pseudo-labeling stage, I still have questions because I felt that some comparisons or curves were missing from the figures (see “Questions”).

Overall, I think this is a valuable contribution for neuroscience / behavior analyses, where annotation is expensive and often inconsistent. However, I do have some doubts regarding the results and I would like clarification before I can fully judge impact (see “Questions” and “Weaknesses”). I am scoring this paper a 4 for now, and I will actively participate in the discussion, if the authors address these questions clearly,  I am inclined to raise my score.

**Strengths:**

The paper tackles an important problem in neuroscience and behavior analysis. The experiments cover the claims made in the paper (the rest is in questions).

**Weaknesses:**

- I find the use of the term “distilled student” somewhat confusing. As I understand it, the pipeline is: (1) train multiple models $f_i$
 , (2) aggregate their predictions across views, (3) run the proposed mvEKS to get temporally consistent keypoints and uncertainty, (4) select only high-confidence frames and remove duplicates with PCA clustering, and (5) treat those frames as pseudo-labels and retrain a single model on GT + pseudo-labels. This looks more like pseudo-labeling with uncertainty filtering and dataset expansion than classical knowledge distillation (where a student is explicitly trained to match a teacher’s outputs/soft targets). That said, if “distillation” is the accepted terminology in this subcommunity, feel free to clarify and ignore this comment.
- I found notation and labels confusing, I put all this into Questions.

**Questions:**

- Figure 3. It looks like the plot does not include the baseline mvEKS variant for pixel error. Could the authors include that for completeness?
- Figure 4. The legend is hard to interpret. For example, does “MVT++ + EKS distilled + Geometric Consistency” use the proposed variance-inflated mvEKS, or the baseline mvEKS? Please make this explicit.
Related to that: because proposed mvEKS + variance inflation seem like core contributions, it would really help to show separate curves for (a) MVT++ + mvEKS (the baseline), (b) MVT++ + mvEKS + variance inflation, (c) MVT++ + mvEKS + variance inflation distilled, and (d) MVT++ + mvEKS  (the baseline) distilled.
- Figures 9, 10, 11, and 16. I did not see results for the mouse dataset in these figures. Why the results were omitted?
- Figure 17. Appendix A suggests that each dataset has <500 labeled frames, but Figure 17 compares “MVT++”, “MVT++ (1000 frames)”, and “MVT++ (3000 frames)”. Are these curves matched for total training compute / number of optimization steps, or are the larger-data models also trained longer? If it’s the latter, could the authors normalize for wall-clock or epochs to make the comparison fair?
- Figure 15. It would help to label clearly which line corresponds to baseline mvEKS and which line corresponds to mvEKS with variance inflation. Also, what is the difference between “Ensemble Median + Anipose” and just “Ensemble Median”? Is Anipose only doing triangulation / post-processing in that comparison?
- Terms mvEKS and EKS are used interchangeably, for example:
> The linear multi-view Ensemble Kalman Smoother (mvEKS), introduced in Biderman et al. (2024), leverages multi-view ...

In the same paragraph authors uses EKS instead. The same is applicable to labels in all the figures. I find it confusing. Could authors clarify if mvEKS and EKS describe the same method that was proposed in Biderman et al. (2024)?

---

> ### Author Response · Authors · 2025-11-20
>
> We thank the reviewer for their thorough feedback, and hope that our answers clarify some of the outstanding questions.
>
> **Distillation terminology.**
> The reviewer has correctly understood our pipeline and accurately characterizes it as "pseudo-labeling with uncertainty filtering and dataset expansion"—this is indeed an apt description of our approach. We appreciate this clear articulation.
> As we noted in our response to reviewer vSia, we view the fundamental goal of distillation as transferring learned capabilities from a high-performing but computationally expensive model to a more efficient one. Our EKS-based approach achieves this by using an ensemble of models with temporal and geometric post-processing (computationally expensive) to generate refined pseudo-labels that train a single efficient model.
>
> However, we acknowledge that our use of the term "distillation" may cause confusion given its strong association with teacher-student frameworks in the machine learning literature. To address this terminological concern, we will update the manuscript to use more precise language such as "pseudo-label based distillation" or "EKS-guided pseudo-labeling" to clearly distinguish our approach from conventional distillation methods.
>
> **mvEKS baseline without variance inflation.**
> We thank the reviewer for bringing this to our attention, this was an oversight on our part. We have now included comprehensive results comparing mvEKS with and without variance inflation across all three datasets in Appendix F.5. These results demonstrate the contribution of our variance inflation strategy and provide the complete experimental context that was missing from our initial submission.
>
> **Omitted mouse results.**
> Fig. 9 (3D augmentations), Fig. 10 (3D loss), Fig. 11 (3D loss) and Fig. 16 (nonlinear EKS) all present results for components that require camera calibration information. The mirror-mouse dataset lacks calibration data and is therefore excluded from these analyses. However, this actually demonstrates an important strength of our framework: our approach is designed to be modular and adaptable to diverse experimental setups with varying levels of available information. We will add explicit clarification in the figure legends explaining why the mirror-mouse dataset is absent from these analyses.
>
> **Distilled model training clarification.**
> The reviewer is correct that each dataset contains fewer than 500 labeled frames. We apologize for the confusion in Fig. 17's legend—the numbers in parentheses refer to the number of pseudo-labels generated by our EKS process, not the number of ground truth labels. All models are trained with exactly 200 ground truth labeled frames, which remains constant across all experiments. We will clarify the legend by changing "N frames" to "N pseudo-labels" to eliminate this confusion and ensure readers understand that we are showing the effect of varying the quantity of pseudo-labeled data while keeping the supervised data fixed.
> To confirm the reviewer's second point: the curves in Fig. 17 are indeed matched for training iterations, with each model (whether using pseudo-labels or not) trained for exactly 5000 iterations. This ensures a fair comparison of convergence behavior and final performance across different pseudo-labeling configurations.
>
> We will update the figure legend and caption to make both of these experimental details explicit in the final manuscript.
>
> **Fig. 15 clarification.**
> We again thank the reviewer for identifying areas where our figures can be improved for clarity.
>
> Regarding the first comment, the baseline mvEKS corresponds to the green line, while mvEKS with variance inflation is shown as the purple line. While these lines frequently overlap due to similar mean predictions, they do diverge in certain regions, notably around frame 300 as highlighted in the figure. The more substantial difference between these methods appears in their uncertainty estimates: the purple line (with variance inflation) typically displays wider error bands, reflecting our method's principled approach to inflating uncertainty when cross-view predictions are inconsistent.
>
> For the second point, could you clarify which figure you are referencing? You mention Anipose, but Figure 15 does not include any Anipose comparisons. If you are referring to a different figure, we would be happy to address your concern once we understand the specific location of the issue.
>
> **EKS/mvEKS.**
> We apologize for the confusion. Every reference to EKS is actually to mvEKS, and we should not have used these terms interchangeably. We will change all references from “EKS” to “mvEKS” throughout the manuscript in the final version.
>
> **Work in progress.**
> We are still working on responding to the following comments:
> * Fig 4 legend interpretability and additional ablations

---

### Official Review · Reviewer_vSia · 2025-10-31

**Soundness:** 1
**Presentation:** 3
**Contribution:** 2
**Rating:** 2
**Confidence:** 5

**Summary:**

The paper proposes a framework for data-efficient multi-view animal pose estimation that integrates: (1) A multi-view transformer (MVT) that processes all camera views jointly, introducing a patch masking scheme to encourage cross-view consistency. (2) A 3D augmentation and triangulation loss for calibrated setups, improving geometric consistency. (3) An enhanced variance-inflated, nonlinear Ensemble Kalman Smoother (EKS) for post-processing and uncertainty calibration. (4) A pseudo-label distillation pipeline, using EKS to generate high-quality labels from unlabeled frames to retrain a single efficient model.
The framework is tested on three datasets (Fly-Anipose, Treadmill Mouse, and Chickadee), showing improvements over single-view ViTs and some multi-view baselines.

**Strengths:**

### **Strengths:**
- A clear motivation: data-efficient animal pose estimation is a relevant challenge.
- The paper proposes a consistent framework that works across both calibrated and uncalibrated setups.
- The variance-inflated EKS for uncertainty refinement is an interesting and potentially valuable extension.
- Demonstrates integration of early-fusion transformers with geometric consistency losses, which could inspire further research.

**Weaknesses:**

### **Weaknesses**

* Lack of quantitative table and ablations to support claims.
* The distillation is mischaracterized; it is pseudo-labelling, not teacher–student distillation.
* No analysis of occlusion handling, despite claiming improvements.
* Masking and augmentation strategies are introduced without proper experimental justification or hyperparameter analysis.
* The paper uses outdated pretraining (DINO); comparison with DINOv2 or MAE-based ViTs is missing.
* No computational efficiency metrics (runtime, model size, speed).
* Terminology and vocabulary sometimes confusing and nonstandard ("3D augmentation *scheme*", "patch masking *scheme*")
* Missing or unclear equations (3D loss, masking formulation).
* A full related work section about MAE would be required, right now all the related work about this domain is overlooked even though it constitutes one of the main authors' claimed contribution.

**Questions:**

### **Questions**

1. How does the model handle occlusions?
2. Why is DINO used rather than DINOv2 (or even v3) or MAE, which are known to produce stronger representations?
3. How was the 10–50% masking ratio chosen (l.203)? How was the masking strategy (l.200) chosen? Any ablation results?
4. Can the authors clarify whether EKS-based pseudo-labelling is equivalent to a true distillation process?
5. How is the 3D keypoint loss implemented mathematically? Please include the full equation.
6. Are there computation or inference benchmarks to show efficiency gains of the distilled model?
7. 3D augmentation is not very clear, at least I'm not sure I understood what was performed. A figure might be more self-explanatory.
8. l.161: Is the positional encoding $p_i$ a contribution or does it already exist? Please cite accordingly. Is the learnable view encoding $v_i$ a contribution or does it already exist? Please cite accordingly.

The paper combines several known ideas (multi-view ViTs, pseudo-labelling, EKS) into a coherent framework, but lacks depth in quantitative validation, novelty, and clarity. With clearer definitions, full ablations, and stronger baselines (e.g., DINOv2, MAE), it could reach the acceptance threshold. Based on the answers to my questions and other reviewer's judgment, I am open to increase my score.

---

> ### Author Response · Authors · 2025-11-20
>
> We thank the reviewer for their thorough feedback, and hope that our answers clarify some of the outstanding questions.
>
> **Lack of quantitative table and ablations to support claims.**
> We have attempted to provide substantial quantitative results to support our claims, and we note that we use the pixel error vs ensemble standard deviation plots (as in Fig. 2c) instead of tables, as these plots provide a much richer picture of model performance. For clarity we note here the ablations we have performed:
> * Single-view vs multi-view patch processing: Fig. 2, 5
> * View masking vs patch masking: Fig. 7
> * 2D vs 3D augmentations: Fig. 9
> * 3D loss hyperparameter: Fig. 10
> * 3D loss, patch masking: Fig. 11
> * Number of training frames: Fig. 12
> * Post-processing: Fig. 3, 16
> * Pseudo-label strategy: Fig. 4, 17
>
> We acknowledge that  this does not represent a comprehensive list of all possible ablations; if there are any ablations in particular the reviewer thinks are critical we are happy to discuss these further. Also, we would appreciate it if the reviewer could highlight any specific claims in the paper they feel are not quantitatively supported.
>
> **Distillation mischaracterization.**
> We did not intend to mischaracterize the approach we are taking, as “distillation” can also refer to the use of pseudo-labels (see, for example, the Distil-Whisper paper, Gandhi et al. 2023). However, as another reviewer also mentioned their confusion with our use of the term “distillation”, we will clarify our use of this term throughout the paper to explicitly state we are using “pseudo-label distillation” (or just “pseudo-labels”).
>
> We maintain that our EKS-based pseudo-labeling constitutes a legitimate form of knowledge distillation, though we acknowledge it differs from conventional teacher-student frameworks. The fundamental principle of distillation—transferring learned capabilities from a high-performing but computationally expensive model to a more efficient one—is preserved in our approach. Our EKS process can be conceptualized as a "teacher" that operates as a smoothed mixture of experts: it combines predictions from multiple ensemble members with temporal smoothing and geometric consistency constraints. The single MVT model serves as the "student" that learns to directly produce these refined predictions. This achieves the core distillation objective: the final MVT model delivers inference that is N times more computationally efficient compared to the EKS ensemble with N members, while maintaining comparable accuracy (Fig. 4).
>
> We are happy to elaborate on any specific aspects of this framework.

---

> ### Author Response · Authors · 2025-11-20
>
> **Masking and augmentation strategies.**
> Our masking strategy can be implemented in multiple ways, and we explored two primary variants: masking random patches across images (patch masking) and masking all patches within entire images (view masking). Our initial comparison showed patch masking outperformed view masking with more stable training dynamics (Figures 6-7).
>
> We acknowledge the reviewer's point regarding hyperparameter analysis and have conducted a systematic study varying the patch mask ratio (0.1, 0.25, 0.5, 0.75) and masking start iteration (0, 700, 1000, 2000). The complete results are provided in Appendix F.3. As expected, performance varies across hyperparameter settings, with optimal values showing some dataset dependence. Importantly, our original hyperparameter choices (ratio=0.5, start iteration=700) demonstrate consistent and reasonably strong performance across all evaluated datasets, suggesting these values provide a robust default configuration. (Having robust defaults that work across datasets is especially relevant for us, as we strive to make tools that work out of the box for experimental labs with as little fine-tuning as possible.)
>
> Regarding 3D augmentations, we have not yet completed a comprehensive hyperparameter analysis, though we plan to include this if time permits. However, we did compare our 3D augmentation approach against standard 2D augmentations (evaluated without the 3D loss component that requires 3D augmentations) and found comparable performance with our initial hyperparameter settings (Fig. 9). This suggests our 3D augmentation strategy is at minimum competitive with established 2D approaches while enabling the additional benefits of 3D geometric supervision.
>
> We will ensure the final manuscript includes more detailed discussion of hyperparameter sensitivity and provides guidance for practitioners implementing our approach.
>
> **Outdated pretraining.**
> We agree that a comparison with DINOv2 would be valuable and are currently implementing this. We note that our initial submission already included results using an MAE-pretrained backbone (ViT/B-ImageNet, Fig. 5), demonstrating our framework's compatibility with different pretrained backbones.
>
> Because of the rapid progress in foundation models for computer vision, our technical contributions are designed to be backbone-agnostic. The core contributions of our work—multi-view processing, patch masking, 3D loss, mvEKS post-processing, and pseudo-labeling—are independent of the underlying vision transformer backbone. This modularity means that as stronger pre-trained models like DINOv2 become available, they can be readily integrated into our framework. We expect that incorporating DINOv2 or other advanced backbones would provide complementary improvements to our multi-view approach, potentially leading to even stronger performance across the evaluated datasets.

---

> ### Author Response · Authors · 2025-11-20
>
> **Computational efficiency.**
> We appreciate the suggestion to benchmark computational efficiency and agree these metrics are crucial for practical adoption. We conducted comprehensive efficiency tests on a standard L4 GPU (24 GB memory) across model variants (ViT-S, ViT-B) and architectural choices (single-view vs. multi-view processing, with/without patch masking and 3D loss). Complete results are provided in Appendix F.4. Below we list some of the key findings:
>
> Training speeds:
> * ViT-S single-view processing achieves the fastest training speeds due to independent view processing, while ViT-B multi-view represents the most computationally intensive configuration due to joint processing of multiple views through the larger architecture.
> * Patch masking introduces negligible training overhead
> * 3D loss contributes a modest increase in batch processing due to additional geometric computations
>
> Inference speeds:
> * Performance ranges from 50-100 FPS for ViT-S single-view (fastest) to 10-20 FPS for ViT-B multi-view (slowest)
> * ViT-S multi-view and ViT-B single-view achieve intermediate performance at approximately 25 FPS
> * ViT-B multi-view is the slowest, at 10-20 FPS
> * Patch masking and 3D losses are only used for training, and hence do not impact inference speed
>
> We have not yet pursued optimization strategies and believe significant efficiency improvements are achievable through standard acceleration techniques if deployment requirements necessitate faster processing.
>
> **Non-standard vocabulary.**
> We regularly use “scheme”, “strategy”, and “technique” interchangeably to reference particular features of our work. We are not aware of any standard vocabulary here, but if the reviewer has suggestions for how to make this clearer we are certainly open to revising the text.
>
> **Missing equations.**
> We agree with the reviewer that explicit equations for the 3D loss and patch masking will add clarity to our work. We have added these in Appendix F.1 (3D loss) and Appendix F.2 (patch masking).
>
> **Masked autoencoding related work.**
> While we acknowledge inspiration from MAE, our patch masking approach differs fundamentally from masked autoencoding in both objective and implementation, and we do not feel an entire Related Work section on MAE is warranted.
> Our method does not perform masked autoencoding: we do not reconstruct input patches or learn representations through reconstruction tasks. Instead, we use patch masking as a form of structured data augmentation that encourages cross-view reasoning for keypoint heatmap prediction. As stated in our initial submission (L199): “Rather than dropping tokens before encoding as in He et al. (2022), we mask patches directly in the input image before patchification, which is more similar to an extreme form of data augmentation mimicking frequent occlusions” than the token-dropping reconstruction paradigm of MAE. We hope the mathematical formulation we provide in Appendix F.2 further clarifies these distinctions.
>
> We will certainly add relevant references to masking techniques in computer vision and search the literature for additional examples of masking strategies beyond autoencoding applications. However, positioning our work primarily within the MAE literature would misrepresent our technical contribution and could mislead readers about the nature of our approach.
>
> **Positional and view embeddings.**
> The position encoding in L161 is not a contribution, this is a standard component of vision transformers; we will clarify this with a reference in L157. We also do not consider the learnable view embedding a contribution, as our work is a variation (though not exactly the same) of implementations in previous work. For example, Ma et al., 2021 and Zhou et al., 2023 introduced view identifiers in late-fusion or hierarchical settings, and we will add these references to the final text (around L161).
>
> **Work in progress.**
> We are still working on responding to the following comments:
> * Occlusion handling
> * DINOv2 backbone
> * Clarification of 3D augmentations

---

> > ### Comment · Reviewer_vSia · 2025-11-25
> >
> > Thank you for the detailed response. Before evaluating the additional results the authors plan to provide (occlusion handling, DINOv2 backbone, and a clearer explanation of the 3D augmentation pipeline), I would like to comment on the points already addressed in the current answer.
> >
> > **Please highlight changes in the manuscript.**
> > It is difficult to assess what has been modified based on the rebuttal text alone. I strongly encourage the authors to visually highlight all updates in the PDF, ideally using a different color for newly added or revised content. This would ensure that reviewers can verify which issues have been addressed (and where) and which ones remain open.
> >
> > **Distillation terminology.**
> > I still disagree with the statement that “EKS-based pseudo-labelling constitutes a legitimate form of knowledge distillation”. The Distil-Whisper example referenced by the authors is indeed a distillation method because it uses (1) a closely related teacher–student architecture where the student is initialized with teacher weights, and (2) a KL divergence loss that explicitly encourages the student to match the teacher’s output distribution. Their loss term \mathcal{L}_{PL} does not play this role (which is an equivalent loss to the one the authors are using I believe). My original point remains unchanged: the method generates pseudo ground truth labels from a mixture-of-experts (and an additional filtering technique) and then trains a model on those pseudo labels. This aligns more closely with the Multi-Model regime described in [Kage et al.](https://arxiv.org/abs/2408.07221) (2024). I recommend adopting terminology that reflects this more accurately.
> >
> > **Masking and augmentation strategies.**
> > The additional analysis is helpful and clarifies some of the design choices. Thank you for explaining this in more details.
> >
> > **Outdated pretraining.**
> > I understand that tracking the rapid progress of foundation models is challenging. However, DINOv2 has been available for more than two years and is widely used as a default backbone in recent pose estimation work because it does not require architectural modifications. I still believe that including at least a minimal DINOv2 baseline is necessary to judge whether the proposed framework is competitive.
> >
> > **Computational efficiency.**
> > Thank you for the additional information. This section is clearer now.
> >
> > **Patch masking and its motivation.**
> > Thank you for the added details. I did not fully grasp this distinction when reading the original submission. However, the following statement remains unclear:
> >
> > > “By masking patches directly in pixel space before patchification, we force the transformer to reason about occluded or missing regions using complementary cues from other views.”
> >
> > This is a strong motivation, but I am not sure it is empirically demonstrated. Since this masking strategy differs significantly from standard approaches, I believe the paper needs a comparison with conventional token-level masking or at least a discussion supported by evidence. If the primary motivation is simplicity of implementation, this should be stated explicitly. If the authors believe the approach is more principled, then the current manuscript does not provide enough justification. Even if a full MAE section is not included, a short related work section on masking strategies would still be necessary because the current description is incomplete.
> >
> > **3D augmentation pipeline and occlusion handling.**
> > Clarification of the 3D augmentation strategy and explicit evidence of occlusion handling remain required for me to provide a final recommendation. These are central claims of the paper, so they need to be demonstrated more rigorously.

---

> > > ### Author Response · Authors · 2025-11-27
> > >
> > > We thank the reviewer for their prompt response, and we have tried to thoroughly address the remaining questions below.
> > >
> > > **Tracking changes**
> > > All updated text in the manuscript is now blue. We continue to keep most of the new plots and text in Appendix F, to make it easy to locate work performed for the rebuttal (and have consistent figure numbers), but we reference these sections and figures throughout the main text now (also in blue). After the rebuttal period we will incorporate these sections into more appropriate locations within the main text and/or Appendix.
> > >
> > > **Occlusion handling.**
> > > We thank the reviewer for suggesting a targeted occlusion analysis, which directly evaluates one of our central contributions. We designed a systematic experiment where we completely mask an entire view from test instances and evaluate the model's ability to predict keypoints in the occluded view using information from remaining views. This process is repeated across all views and test instances to provide comprehensive occlusion performance metrics.
> > >
> > > The single-view transformer (SVT) serves as our baseline: since it processes views independently, complete view occlusion should result in poor predictions due to lack of access to complementary information. We compare SVT against MVT, MVT with patch masking, and (for calibrated datasets) MVT with 3D loss, and the full MVT++. Any improvement over SVT demonstrates successful utilization of cross-view information to compensate for occlusions.
> > >
> > > Results across all datasets are presented in Figure 22. Our findings strongly support our claims: patch masking provides substantial improvements in occlusion handling, reducing SVT occlusion errors by 71% on Fly-Anipose, 42% on Chickadee, and 31% on Treadmill Mouse. These results demonstrate that our cross-view patch masking strategy effectively teaches the model to leverage complementary information across views.
> > >
> > > The analysis reveals a clear performance hierarchy: MVT alone provides moderate gains over SVT by enabling cross-view reasoning. Adding patch masking yields the largest improvements by explicitly encouraging occlusion robustness within the MVT (i.e. the architecture is capable of effective occlusion handling, but must be trained in particular ways to enable this functionality). The 3D loss provides additional modest gains, while the full MVT++ combination achieves the best overall occlusion handling performance. These results validate our approach and demonstrate meaningful practical benefits for handling the occlusions that are ubiquitous in multi-view behavioral recordings.
> > >
> > > | Dataset | SVT | MVT | MVT+3D | MVT+patch | MVT++ |
> > > |---------|-----|-----|--------|-----------|-------|
> > > | Treadmill Mouse | 105.85 | 104.65 | N/A | **73.20** | N/A |
> > > | Fly-Anipose | 77.27 | 64.24 | 60.48 | 25.22 | **22.34** |
> > > | Chickadee | 42.62 | 41.81 | 40.48 | 23.85 | **22.72** |
> > >
> > > We have also included a new figure, Fig. 23, that shows a qualitative example of how SVT and MVT++ deal with occlusions in the Chickadee dataset.
> > >
> > > Finally, we note that in the response to reviewer q2Cc, we performed an ablation study on the number of camera views for the Fly-Anipose and Chickadee datasets (which each contain 6 views). For this study we chose a random subset of 4 views on which to train the MVT; we also chose a further random subset of 2 views. We trained models across 2, 4, and 6 view datasets, then evaluated all models on test frames from the 2 views included in each subset. This allows us to gauge the effect of additional views when training the MVT. Notably for Fly-Anipose, the animal is head-fixed, which means that many of the keypoints are occluded on most frames for certain views (for example, keypoints close to the abdomen on the right side are always occluded from cameras on the left side). Adding more camera views significantly improves performance in this dataset, and the Chickadee dataset as well, again indicating the MVT has learned to leverage information from other views to produce more accurate predictions.
> > >
> > > **Masking strategies Related Work.**
> > > In order to properly situate our patch masking strategy in the literature, we have added a Related Work section in Appendix F.3, and clarified how our approach differs from existing work. Depending on page constraints we will either put this text in the Related Works section of the main text, or reference an Appendix section from the main text. We believe that this context, along with the quantitative analysis from the previous point, have improved the manuscript by more clearly highlighting this technical contribution.

---

> > > > ### Author Response · Authors · 2025-11-27
> > > >
> > > > **Distillation terminology.**
> > > > We appreciate the reviewer’s insistence on this point, as it was clearly a point of confusion for at least one other reviewer as well (and therefore would be for many readers). We have adopted the “pseudo-label” terminology throughout the paper; changes are highlighted in blue (we are still working to update Fig. 4).
> > > >
> > > > **DINOv2 backbone.**
> > > > We agree with the reviewer that including DINOv2 as a baseline strengthens our evaluation and we have implemented this comparison. We trained both single-view and multi-view transformer models using DINOv2 backbones (3 seeds each), with complete results presented in Appendix F.9 (Figs. 24 and 25). The results reveal interesting dataset and architecture dependencies: ViT-B DINOv2 achieves best performance for both SVT and MVT on the Fly-Anipose and Chickadee datasets, while ViT-S DINO performs best on Treadmill Mouse. These findings highlight that backbone selection can be dataset-dependent, likely reflecting differences in visual complexity, imaging conditions, and the specific pose estimation challenges presented by each experimental setup.
> > > >
> > > > This analysis also provides valuable guidance for practitioners: users implementing our framework should consider evaluating multiple backbone options for their specific datasets to optimize performance. The modularity of our approach ensures that stronger backbones can be readily incorporated as they become available, making our contributions robust to ongoing advances in vision transformer architectures.
> > > >
> > > > Regarding a ViT-S pretrained with MAE, we would like to have included this as a baseline but were not able to find such a pretrained model on Hugging Face, which we used for our experiments. We did find a ViT-B pretrained with MAE (https://huggingface.co/facebook/vit-mae-base), and results for this model are included in the original architecture baselines in Fig. 5.
> > > >
> > > > **Clarification of 3D augmentations.**
> > > > We have created a figure (Fig. 26 in Appendix F.10) that illustrates the 3D geometric component of the augmentations. We also refer the reviewer to Fig. 8 for example augmentations on real frames. If there is any component of the augmentation pipeline that is still unclear (the 2D non-geometric augmentations, or the affine transformation applied to the image after keypoint augmentations), we are happy to further clarify.

---

### Official Review · Reviewer_q2Cc · 2025-11-01

**Soundness:** 2
**Presentation:** 3
**Contribution:** 2
**Rating:** 4
**Confidence:** 4

**Summary:**

This paper presents an uncertainty-aware framework for data-efficient multi-view animal pose estimation, addressing key challenges in quantifying animal behavior for scientific research. The framework has three components: 1. Multi-view Vision Transformer (MVT): An architecture that processes image patches from all camera views simultaneously, enabling early fusion of cross-view information through self-attention. 2. Enhanced Post-processing: An improved Ensemble Kalman Smoother (EKS) featuring a nonlinear variant and a variance inflation technique for better spatiotemporal smoothing and uncertainty calibration. Distillation Pipeline: A method to transfer knowledge from the complex multi-model EKS pipeline into a single, efficient network using high-quality pseudo-labels. The method can work without camera calibration. It is validated on diverse species datasets.

**Strengths:**

1.	The proposed components provide complementary benefits. For example, patch masking and 3D loss address different aspects of robustness ( occlusion handling vs. geometric consistency).
2.	The approach could adapt to different camera settings (with calibration or without calibration), enabling broader application scenarios.
3.	The paper demonstrates superior performance with very limited labeled data.
4.	This paper is easy to read.

**Weaknesses:**

1. The technical contributions seem limited. (1) The patch masking operation is commonly seen in today’s vision transformer studies. (2) nonlinear MvEKS only involves standard camera distortion (with radial and tangent distortion parameters) in the Kalman filter.

2. The complexity of MVT may increase with the number of camera views increasing.

3. Why not directly apply Kalman filter to 3D data x? would the performance degrade compared with applying Kalman filter to 2d keypoints then deducing 3D?

4. For typical triangulation, the accuracy of 3D point estimation would increase given more number of observed views. However, I do not know how the performance of MVT would be affected by the number of camera views. An ablation study would help understand the behavior of MVT.

5. A previous paper named “Triangulation residual loss for data-efficient 3D pose estimation” also addresses data-efficient 3d pose estimation of diverse species. If possible, a comparison would strengthen the paper.

6. The 3D baseline only includes Anipose, limiting the persuasiveness of the experiments. Including more baselines such as selfpose3d or dannce (trained with limited samples) would enhance the experiments.

**Questions:**

See above. Currently the results in the paper are not very impressive for me, as the techniques used are not surperising. The paper writing is good. The integrity of experiments still has space to improve.

---

> ### Author Response · Authors · 2025-11-20
>
> We thank the reviewer for recognizing the flexibility of our approach and specifically the performance benefits when training with few labels. We try to clarify any remaining concerns with our comments below.
>
> **W1: Technical contributions.**
> (1) While patch masking is indeed common in vision transformers, our approach differs fundamentally from existing methods like Masked Autoencoders (MAE). Standard masking schemes reconstruct missing patches within a single view using visible patches from that same view. Our method introduces a novel cross-view masking strategy that uses visible patches across multiple camera views to reconstruct complete heatmaps for all views and keypoints simultaneously. This cross-view formulation is technically distinct (see updated Appendix F.2) and addresses a different challenge: encouraging the model to leverage complementary information across views when patches are occluded in individual cameras. Our ablation study (Fig. 7) demonstrates that attention mechanisms alone, without our masking strategy, achieve inferior performance, confirming that the technical novelty lies in the cross-view information propagation enabled by our masking approach.
>
> (2) We acknowledge that standard camera distortion models are well-established. However, our technical contribution lies in the principled integration of these nonlinear models as fixed observation functions within an Extended Kalman Filter framework for multi-view pose estimation. Specifically, we introduce time-varying observation variances estimated through ensemble variance combined with our novel variance inflation strategy, rather than learning fixed observation variances from data (the standard approach in probabilistic models). This variance inflation method systematically increases ensemble variance for keypoints and frames where 2D predictions show inconsistency across views, providing a principled approach to handle uncertainty in multi-view settings. This represents a meaningful technical advance over standard EKF formulations.
>
> A core contribution of our work lies in the domain-specific adaptation and integration of existing techniques (transformers, Kalman filters) to address the underexplored problem of multi-view pose estimation with limited training data and compute. We believe this type of methodological contribution represents a valuable scientific contribution, particularly given the performance gaps we address.
>
> We will enhance the manuscript to better highlight these technical distinctions and hope the reviewer will consider the value of both our novel technical components and our systematic approach to an important application domain.
>
> **W2: Increasing camera views.**
> We agree that the number of tokens increases with additional camera views. However, our design keeps computational and parameter complexity modest with increasing views. MVT uses a single shared transformer backbone that jointly processes all views, without view-specific branches or pairwise attention modules, so the parameter count remains constant regardless of view number. The primary cost scales approximately quadratically with the number of views (due to the self-attention mechanism) and remains practical for typical experimental setups (2-6 cameras). Empirically, we used the same architecture and hyperparameters for two-view (Treadmill Mouse) and six-view (Fly-Anipose, Chickadee) datasets with no instability or memory issues with a relatively small GPU (24 GB), confirming the framework’s scalability for realistic scenarios. We believe this tradeoff between modest computational increase and substantial accuracy/robustness gains is a key strength of the MVT design.

---

> ### Author Response · Authors · 2025-11-20
>
> **W3: Kalman filter on 3D data.**
> Our mvEKS provides a principled framework that jointly addresses two complementary aspects of multi-view pose estimation: **temporal smoothing** of noisy observations across time, and **geometric consistency** enforcement across camera views. Crucially, the mvEKS simultaneously leverages both spatial consistency constraints across views and temporal smoothness constraints over time, while adaptively weighting each view based on estimated uncertainty.
>
> Applying Kalman smoothing directly to 3D coordinates fundamentally decouple this joint optimization into two sequential steps: (1) fusing 2D predictions across views to obtain 3D coordinates (spatial consistency), followed by (2) temporal smoothing of these 3D estimates. This sequential approach cannot exploit the complementary nature of spatial and temporal constraints. Specifically, our joint formulation enables the temporal smoothness prior to inform spatial fusion in cases where cross-view consistency is poor (e.g., due to occlusions or detection failures), while simultaneously allowing spatial information from multiple views to improve temporal tracking when individual view predictions are unreliable. For instance, when 2D detections across views are spatially inconsistent, the temporal dynamics model can guide the inference toward kinematically plausible solutions. Conversely, when temporal predictions are uncertain (e.g., during rapid motion), robust spatial information from multiple views can stabilize the tracking.
> Applying Kalman smoothing independently to 2D points from each view before 3D fusion would similarly fail to leverage cross-view information during the temporal smoothing process, potentially missing opportunities to improve tracking robustness when individual views are compromised.
>
> Our joint formulation thus represents a meaningful technical advance over sequential approaches by exploiting both spatial and temporal constraints in multi-view settings. We will clarify these technical distinctions and their practical implications more thoroughly in the final manuscript.
>
> **W5: Triangulation residual loss.**
> We appreciate the reviewer for bringing this interesting and relevant work to our attention. The Triangulation Residual loss represents an elegant approach for leveraging unlabeled multi-view data to improve pose estimation networks, particularly in low-data regimes. We view the TR loss as conceptually similar to the multi-view PCA loss from Lightning Pose, which demonstrated that unsupervised multi-view consistency losses can effectively complement supervised model improvements.
>
> We believe the same principle could apply to the MVT and TR loss: MVT focuses on cross-view attention mechanisms and joint feature extraction, while TR loss provides an unsupervised consistency objective. These approaches could easily be integrated, with TR loss serving as an additional training objective alongside our existing components (cross-view masking, 3D supervision).
>
> The integration potential is particularly promising from an architectural perspective. The original TR loss implementation operates on CNN-based backbones that process views independently before applying geometric consistency constraints. Implementing TR loss on top of our MVT backbone would provide several advantages: first, the TR loss would benefit from stronger initial 2D predictions generated by our cross-view attention mechanism. Second, and more importantly, large TR loss values could provide an additional supervisory signal to strengthen the cross-view feature extraction capabilities of the MVT backbone in ways that are impossible when processing views in isolation. We therefore see MVT and TR loss as addressing different but complementary aspects of the multi-view pose estimation challenge, with significant potential for beneficial integration.
>
> We will add this reference to our Related Work section and acknowledge TR loss as a promising complementary technique. Exploring the integration of TR loss with MVT represents an exciting direction for future work that could further improve performance, particularly in scenarios with abundant unlabeled multi-view data.
>
> **Work in progress.**
> We are still working on responding to the following comments:
> * W4: Performance as a function of camera views
> * W6: DANNCE baseline

---

> > ### Comment · Reviewer_q2Cc · 2025-11-26
> >
> > I appreciate the efforts of the authors to make some points more clarified.
> >
> > I think "domain-specific adaptation and integration of existing techniques" is important, however, ICLR is a venue focusing on the technical advancements. I think this paper is a bit limited in technical contributions. Therefore I maintain my initial rating.

---

> > > ### Author Response · Authors · 2025-11-27
> > >
> > > **W1: Technical contributions.**
> > > We respectfully disagree with the characterization that our work lacks important technical advances. Our contributions represent meaningful innovations that address fundamental challenges in multi-view pose estimation:
> > > - multi-view transformer architecture: We are not aware of prior work that employs self-attention mechanisms in vision transformers to directly process multiple camera views at the image level for pose estimation. Plenty of works utilize late-fusion strategies where image features are extracted view-by-view, and then a transformer processes all featuremaps, heatmaps, or predicted keypoints simultaneously (see Related Works). This approach fails to leverage cross-view attention for the feature extraction stage, which we address with our MVT. If the reviewer is aware of similar prior work, we would greatly appreciate a reference.
> > > - cross-view patch masking: Our masking strategy is technically distinct from MAE and other reconstruction-based approaches. Rather than reconstructing masked patches, we use masking to encourage cross-view reasoning for keypoint prediction (see new Related Work section in Appendix F.3). Our new analysis (Fig. 22) demonstrates the substantial benefits of this approach for handling occlusions. We have also included a new Related Works section in Appendix F.3 to better situate our work in the literature.
> > > - Variance inflation for mvEKS: Our systematic comparison between standard and variance-inflated EKS (updated Fig. 20) shows substantial improvements, particularly for challenging keypoints, demonstrating the value of principled uncertainty estimation in multi-view settings.
> > >
> > > These technical contributions address a scientifically important but underserved domain: pose estimation with severely limited labeled data. Training effective models with only 200 (or fewer) labeled instances represents a fundamentally different challenge from human pose estimation, where millions of labels are available. This limited-label scenario is pervasive across scientific disciplines where expert annotation is expensive and time-consuming.
> > >
> > > Our work demonstrates how computer vision techniques can be systematically adapted with domain-specific innovations to create practical tools for scientific applications. The combination of our technical advances produces state-of-the-art results in this challenging regime, representing meaningful progress for the broader scientific community.
> > >
> > > **W4: Performance as a function of camera views.**
> > > The reviewer's intuition that accuracy increases with the number of observed views is correct. This relationship has been demonstrated previously with triangulation and other multi-view methods in multiple papers, including, strikingly, in the OpenMonkeyStudio paper (Bala et al., 2020), where the authors found that pose estimation performance continued to improve across their 62 camera views without saturating.
> > >
> > > To address this comment, we trained MVT models on the Fly-Anipose and Chickadee datasets (both 6 views) using either 2, 4, or 6 views (3 random seeds each). We selected the 2 views as a random subset of the 4 randomly selected views, and evaluated all models on 2D keypoint predictions from the same shared 2-view subset. This design isolates the benefit of additional views during training, since all models are evaluated on identical camera views.
> > >
> > > We find that including more views during training leads to better performance on both datasets, with larger improvements for Fly-Anipose. The stronger effect on Fly-Anipose likely reflects the head-fixed setup, where certain keypoints are consistently occluded in specific camera views. The MVT can leverage information from additional views during training to improve predictions even on the held-constant evaluation views (See Appendix F.8 for a rigorous quantification of this statement). Results are shown in the updated Appendix F.7.
> > >
> > > **Work in progress.**
> > > We are still working on responding to the following comments:
> > > * W6: DANNCE baseline: we are training models now.

---

> > > > ### Author Response · Authors · 2025-12-02
> > > >
> > > > We implemented the DANNCE baseline for both calibrated datasets (Fly-Anipose and Chickadee), as camera calibration is required for DANNCE. We contacted the original authors to verify our training configuration given the challenges of training DANNCE with limited labeled data.
> > > >
> > > > For Fly-Anipose, DANNCE achieves substantially higher pixel error (23.13 ± 0.11, mean ± SEM) compared to our MVT++ (8.55 ± 0.03). Complete results and example predictions are shown in Fig. 27, demonstrating that while DANNCE learns overall pose structure, it struggles with precision in our limited-data regime.
> > > >
> > > > The Chickadee dataset presents a particularly informative case, as it was included in the original DANNCE paper. However, that work fine-tuned a network pretrained on tens of thousands of labeled rat frames, whereas we trained from a randomly initialized network. We were unable to achieve meaningful convergence on Chickadee data without this extensive pretraining, highlighting a key limitation of the DANNCE approach.
> > > >
> > > > This comparison underscores a fundamental advantage of our method: MVT models leverage standard pretrained vision transformer backbones (DINO) that provide strong general visual representations, eliminating the need for domain-specific 3D pose datasets. This makes our approach considerably more practical for new species or experimental setups where extensive 3D pose data is unavailable. The performance gap between DANNCE and MVT++ on Fly-Anipose (23.13 vs 8.55 pixel error) demonstrates the substantial benefits of our approach in realistic limited-data scenarios.

---

### Author Response · Authors · 2025-11-20

We sincerely thank the reviewers for their thoughtful evaluations and constructive feedback. We believe that addressing these points has strengthened our submission significantly.

Below we address the comments of each individual reviewer. We have organized our responses into three categories:

1. **Minor revisions**: For straightforward concerns (e.g., adding missing references, clarifying writing), we acknowledge the feedback and indicate the changes we will make in the final manuscript.

2. **Substantive responses with new results**: For more in-depth comments requiring additional experiments, we have added a new section to the appendix of our manuscript (Appendix F). We will post initial conclusions in our text responses here and direct you to specific sections of the updated PDF for detailed results and visualizations. We will continue to update this manuscript as we complete additional analyses.

3. **Work in progress**: For comments we are still actively addressing, we list these at the end of each reviewer response and will post follow-up comments as new results become available.

We have already uploaded an initial manuscript revision to facilitate discussion. We plan to update this document iteratively over the coming weeks and will notify you when significant new results are added.

We hope this structure will assist in your evaluation and look forward to continuing discussions with each of you.

---

> ### Author Response · Authors · 2025-12-02
> **Summary of updates**
>
> We sincerely thank the reviewers again for their time and constructive feedback. We regret that the discussion period ended prematurely, and have worked hard to thoroughly address all remaining comments and questions.
>
> **Reviewers acknowledged several strengths of our work:**
> * The problem of pose estimation with limited data is well-motivated, and the proposed methods provide superior performance in this challenging regime.
> * Our framework demonstrates valuable flexibility in leveraging camera calibration information when available.
> * The paper is well-written and ideas clearly presented.
>
> Responding to reviewer feedback has significantly strengthened our submission through additional baselines, ablations, and clarifying analyses. Below we summarize the key updates, highlighted in blue text in the revised manuscript.
>
> **New Baselines**
> * **DINOv2**: Implemented DINOv2 (Small and Base) backbones with comprehensive evaluation across single-view and mulit-view transformers. Results showed dataset-dependent performance that did not consistently justify replacing our DINO backbone, but this backbone (and others) can be considered in the future (Appendix F.9).
> * **DANNCE**: Made extensive efforts to implement DANNCE (Dunn et al., 2021) as a baseline, including consultation with the original authors (note the code base has not been maintained for several years). However, reasonable performance could not be achieved, likely due to the substantial difference in data regimes: DANNCE requires tens of thousands of labeled frames while our approach excels with only hundreds.
>
> **Ablations**
> * **Patch masking hyperparameters**: Systematic analysis of mask ratio and start iteration parameters validates our initial hyperparameter choices while demonstrating reasonable sensitivity (Appendix F.4).
> * **Variance inflation**: Quantitative demonstration of EKS with and without our novel variance inflation (VI) technique, highlighting VI’s benefits and addressing a key technical contribution (Appendix F.6).
>
> **Additional Improvements**
> * **Occlusion handling**: Direct evaluation of cross-view reasoning capabilities, demonstrating that patch masking reduces occlusion errors by 31-71% across datasets (Appendix F.8).
> * **Computational efficiency**: Benchmarking of training and inference performance, showing practical frame rates of 10-20 FPS for our most capable models (Appendix F.5).
> * **Multi-camera scaling**: Analysis demonstrating improved performance with increasing camera views, complementing our occlusion analysis (Appendix F.7).
> * **Mathematical formulations**: Added detailed equations for 3D loss (Appendix F.1) and patch masking (Appendix F.2).
> * **References**: Added missing citations and included a new Related Work section on masking strategies (Appendix F.3).
>
> The new figures and analyses are currently organized in “Appendix F: Rebuttal Appendix” to maintain consistent numbering throughout the review process. For the camera-ready version, we will integrate these figures and accompanying text into the appropriate sections of the main manuscript and standard appendix to ensure optimal organization and readability.

---

### Note · Program_Chairs · 2026-01-17
**Submission Desk Rejected by Program Chairs**

The following references in this submission do not refer to real documents and/or have major errors in bibliographic information:

 Jinwoo Park and Bohyung Kim. How do vision transformers work? insights from spatial masking. In Proceedings of the IEEE/CVF Conference on Computer Vision and Pattern Recognition, 2022. URL